# Identification of endothelial and mesenchymal *FOXF1* enhancers involved in alveolar capillary dysplasia

Guolun Wang [1,2] ✉, Bingqiang Wen[3], Minzhe Guo [1,2], Enhong Li[3], Yufang Zhang[1], Jeffrey A. Whitsett [1,2], Tanya V. Kalin[3] & Vladimir V. Kalinichenko [3,4] ✉

Mutations in the *FOXF1* gene, a key transcriptional regulator of pulmonary vascular development, cause Alveolar Capillary Dysplasia with Misalignment of Pulmonary Veins, a lethal lung disease affecting newborns and infants. Identification of new *FOXF1* upstream regulatory elements is critical to explain why frequent non-coding *FOXF1* deletions are linked to the disease. Herein, we use multiome single-nuclei RNA and ATAC sequencing of mouse and human patient lungs to identify four conserved endothelial and mesenchymal *FOXF1* enhancers. We demonstrate that endothelial *FOXF1* enhancers are auto-activated, whereas mesenchymal *FOXF1* enhancers are regulated by EBF1 and GLI1. The cell-specificity of *FOXF1* enhancers is validated by disrupting these enhancers in mouse embryonic stem cells using CRISPR/Cpf1 genome editing followed by lineage-tracing of mutant embryonic stem cells in mouse embryos using blastocyst complementation. This study resolves an important clinical question why frequent non-coding *FOXF1* deletions that interfere with endothelial and mesenchymal enhancers can lead to the disease.

The development of the mammalian lung is initiated by complex signal interactions between the endoderm and surrounding mesoderm in the foregut[1]. Lung mesoderm progenitors give rise to both endothelial and mesenchymal cells that provide morphogenetic signals required for epithelial branching and differentiation. FOXF1 is a member of the FOX family of transcription factors that are highly enriched in lung endothelial and mesenchymal cell lineages[2,3]. FOXF1 is required for lung branching morphogenesis and the development of pulmonary vasculature in mice and humans[4]. *Foxf1−/−* mice are embryonic lethal and *Foxf1* haploinsufficiency causes lung hypoplasia with paucity of alveolar capillaries (alveolar capillary dysplasia)[5,6] and inhibits lung repair after injury[7]. Heterozygous deletions and point mutations in the human *FOXF1* gene locus are linked to Alveolar Capillary Dysplasia with Misalignment of

Pulmonary Vein (ACDMPV), a rare congenital disorder which causes respiratory failure in neonates or infants[8].

Recent single-cell RNA sequencing of human and mouse neonatal lung tissues identified two types of alveolar capillary endothelial cells: general capillary cells (gCAP or CAP1) and alveolar capillary cells (aCAP or CAP2). FOXF1 is expressed in aCAPs and gCAPs as well as in multiple mesenchymal cells, including pericytes and fibroblasts[9]. FOXF1 stimulates vascular repair after neonatal hyperoxic lung injury by promoting angiogenesis[10]. FOXF1 + KIT+ gCAPs were recently identified as endothelial progenitor cells that are capable of engraftment into the neonatal lung tissue of ACDMPV mice to promote the formation of new alveolar capillaries[9]. Since many non-coding deletions in or near the *FOXF1* gene locus cause human ACDMPV[8], identification of pathogenic enhancers critical for *FOXF1* gene expression is important

---

[1]Division of Neonatology and Pulmonary Biology, Perinatal Institute, Cincinnati Children's Research Foundation, Cincinnati, OH, USA. [2]Department of Pediatrics, University of Cincinnati College of Medicine, Cincinnati, OH, USA. [3]Phoenix Children's Research Institute, Department of Child Health, University of Arizona, College of Medicine - Phoenix, Phoenix, AZ, USA. [4]Division of Neonatology, Phoenix Children's Hospital, Phoenix, AZ, USA.
✉ e-mail: guolun.wang@cchmc.org; vkalin@arizona.edu

to understand and diagnose the pathobiology of ACDMPV, enabling better genetic screening for ACDMPV which currently relies primarily on exome DNA sequencing.

Genomic organization of the *FOXF1* gene locus, located in a region spanning hundreds of kilobase pairs[11], is highly conserved in mammals. Sequence alignments of non-coding 16q24 deletions in ACDMPV patients identified a shared deletion region (SDR) on chr16:86,212,040–86,287,054 (GRCh37/hg19), located upstream of the *FOXF1* gene[12]. The subsequent genome analysis of DNA sequence from ACDMPV patients sharing overlapping genomic deletions identified a narrower ~15 kb deletion in the *FOXF1* non-coding region (GRCh37/hg19, chr16:86,238,601-86,253,509)[13], which is pathogenic. While these published studies indicate the presence of *FOXF1* regulatory elements critical for ACDMPV pathogenesis, the detailed molecular mechanisms whereby these non-coding regions control *FOXF1* gene expression are unknown.

Single-cell and single-nucleus RNA sequencing provides an even more detailed understanding of the genes and pathways that define cell identity during lung morphogenesis. Single-nuclei ATAC sequencing provided precise information about cell-specific cis-regulatory elements[14]. The 10X multiome (GEX + ATAC) is an emerging technology that enables simultaneous profiling of both transcriptome and epigenome at single-cell resolution.

In the present study, we performed the single-nuclei multiome sequencing of FOXF1-expressing cells to examine gene expression and chromatin accessibility of non-coding *FOXF1* regulatory sequences in mouse and human ACDMPV lungs. Our studies identified four evolutionarily conserved upstream enhancers in *FOXF1* gene locus (FOXF1 Expression in the Lung (FEL) – 1, 2, 3 and 4) and demonstrated that these FEL regulatory elements mediate cell-specific *FOXF1* expression in pulmonary endothelial cells and mesenchymal cells. Human ACDMPV cases with non-coding *FOXF1* deletions were associated with loss of one or more FEL regulatory elements. We also identified transcription factors regulating activities of these FEL enhancers and validated cell specificity of FEL1 and FEL4 using CRISPR/Cpf1 genome editing in pluripotent embryonic stem cells (ESCs) followed by in vivo tracing of mutant ESCs in mouse embryonic lung tissue using blastocyst complementation. Our studies demonstrated that the *FOXF1* enhancers play critical roles in pathogenesis of ACDMPV and can be useful for future genetic diagnosis of the disorder.

## Results

### Identification of FOXF1-expressing cells in the lung
To identify critical regulatory regions in the *Foxf1* promoter, we profiled the transcriptome and epigenome of FOXF1-expressing cells using the multiome (GEX + ATAC) sequencing technology. Since the *Foxf1-GFP* knock-in reporter faithfully recapitulates the expression pattern of the endogenous *Foxf1* gene[15], the sequencing library was prepared from GFP-positive lung cells isolated by FACS sorting from six *Foxf1-GFP* transgenic mice at E18.5 lungs (Fig. 1a and Supplementary Fig. 1a). E18.5 was chosen for multiome sequencing because differentiation of aCAPs occurs around E17.5-E18.5 and continues after birth in the mouse lung. After initial processing of the sequencing output by R package Seurat[16] and Signac[17], 5973 nuclei passed quality controls (Supplementary Fig. 1b, c). The 5973 nuclei were further processed by unsupervised clustering and UMAP embedding with RNA library alone or joint projections with RNA and ATAC library using the weighted nearest neighbor method[18] (Fig. 1b). The cell nuclei were separated into 9 clusters using either RNAseq alone or a joint projection of RNAseq and ATACseq based on gene signatures and epigenetics. Four mesenchymal clusters were identified, including pericytes, matrix fibroblasts, fibroblasts, and myofibroblasts. Five endothelial cell clusters, including CAP1 (gCAP), CAP2 (aCAP), arterial, venous and lymphatic endothelial cells were identified. Representative cell selective markers were identified

using the R package Seurat, and the top 10 transcripts in each cluster were further validated based on the previously published data[19] and the publicly available Lunggens online portal (https://research.cchmc.org/pbge/lunggens/default.html). Consistent with previous studies[20], CAP1 expressed *Ptprb*, *Aplnr* and *Kit*, whereas CAP2 expressed *Ednrb*, *Apln* and *Car4*. (Fig. 1c) Arterial endothelial cells expressed *Gja5* and *Bmx*, while venous cells expressed *Csrp2* and *Vwf*. Lymphatic endothelial cells we identified by *Prox1* and *Mmrn1* transcripts. Specific markers for mesenchymal cell types were consistent with published studies[19,21], in which *Pdgfrb* and *Cspg4* were identified in pericytes, *Mfap4* and *Macf1* in matrix fibroblasts, *Pdgfra* and *Tgfbi* in fibroblasts, and *Acta2* and *Actg2* in myofibroblasts. The matrix fibroblast cell cluster was further sub-divided into two sub-clusters: matrix fibroblast-1 (expressing *Prkg1* and *Adamts17*) and matrix fibroblast-2 (expressing *Igfbp7* and *Egr1*), consistent with published studies[9,19] (Supplementary Fig. 2a, b). Recent single cell RNA sequencing data identified heterogenic structures in the pulmonary capillary network[22]. Our multiome sequencing further revealed that gCAP cell cluster consists of four cell subsets (Supplementary Fig. 2c), all of which express gCAP markers *Kit* and *Gpihbp1* (Supplementary Fig. 2d). In addition to the main sub-cluster 1 with high levels of *Scn7a* and *Sparcl7* mRNAs, we identified the sub-cluster 2 expressing *Tmem100* and *Sema3c* (Supplementary Fig. 2e), genes critical for TGFβ signaling and vascular patterning[23]. The enrichment of *Mki67* and *Top2a* in the sub-cluster 3 suggest that this gCAP population is highly proliferative (Supplementary Fig. 2e). Selective expression of *Adgrg6* and *Fbln5* in sub-cluster 4 (Supplementary Fig. 2e) indicates that these cells are involved in lipid metabolism[24].

Next, we performed integrative analysis of the single-nuclei RNAseq library from the multiome sequencing and publicly available single-cell RNAseq library[15]. Both RNAseq libraries were generated from cells obtained from *Foxf1-GFP* transgenic E18.5 lungs using the same FACS-sorting experimental conditions. Cell clusters from both libraries aligned well (Supplementary Fig. 3a–c). The cell composition and representative cell markers were similar in these two independent RNAseq libraries (Supplementary Fig. 3d, e), demonstrating the reproducibility of cell clustering in the mouse lung.

### Identification of cell type-specific enhancers
Since Single-nuclei ATAC sequencing captures the chromatin accessibility profile of individual cells, we examined the chromatin accessibility in the murine *Foxf1* genomic region. Approximately 56% (3323) of all analyzed nuclei were from endothelial cells, whereas the remaining 44% (2650) nuclei in our snATACseq dataset were from mesenchymal cells. (Fig. 2a). Chromatin accessibility across various lung cell types was examined and 140,580 accessible open chromatin regions from 5973 nuclei were identified. Cell types were first distinguished based on differentially expressed genes and open chromatin regions in each cell type and were further analyzed using the R package Signac[18]. In all cell types examined, most open chromatin sites were enriched in the 5 kb DNA regions flanking the transcription start site across various cell types (Fig. 2b and Supplementary Fig. 1d). Transcription DNA footprinting analysis showed an enrichment of FOXF1-binding sites in CAP1 and CAP2 cells (Fig. 2c), a finding consistent with previous studies showing that FOXF1 activates its own transcription[9]. Since the activity of specific transcription factors (TF) can be predicted by examining their binding motifs[25], we used the differential accessibility test[26] to predict TF pseudoactivity by ChromVAR[26] from snATACseq library and to identify cell types in which the TFs are enriched. The activity of SOX7 was increased in CAP1 cells, whereas the ETS1 activity was higher in arterial endothelial cells (Fig. 2d). ETV6 activity was increased in venous cells, and ELF1 in the lymphatics (Fig. 2d). The open chromatin binding sites for EBF1, TCF7, MEF2c/2d and TCF21, were enriched in pericytes, fibroblasts, myofibroblasts and matrix fibroblasts,

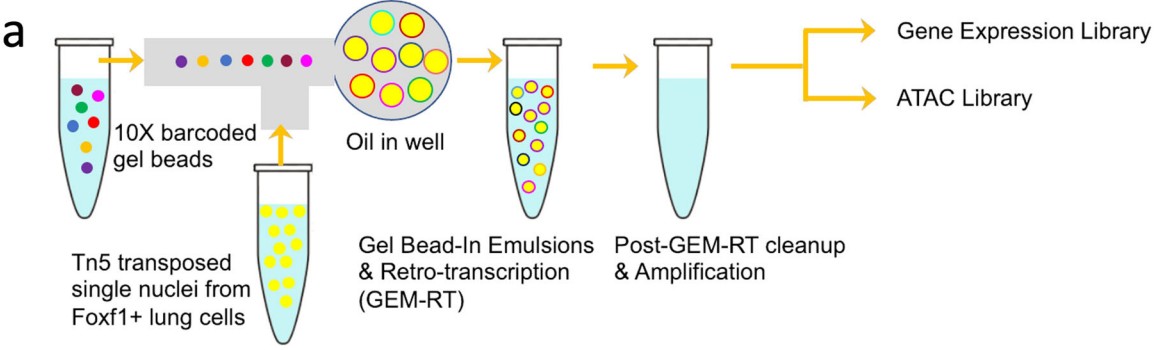

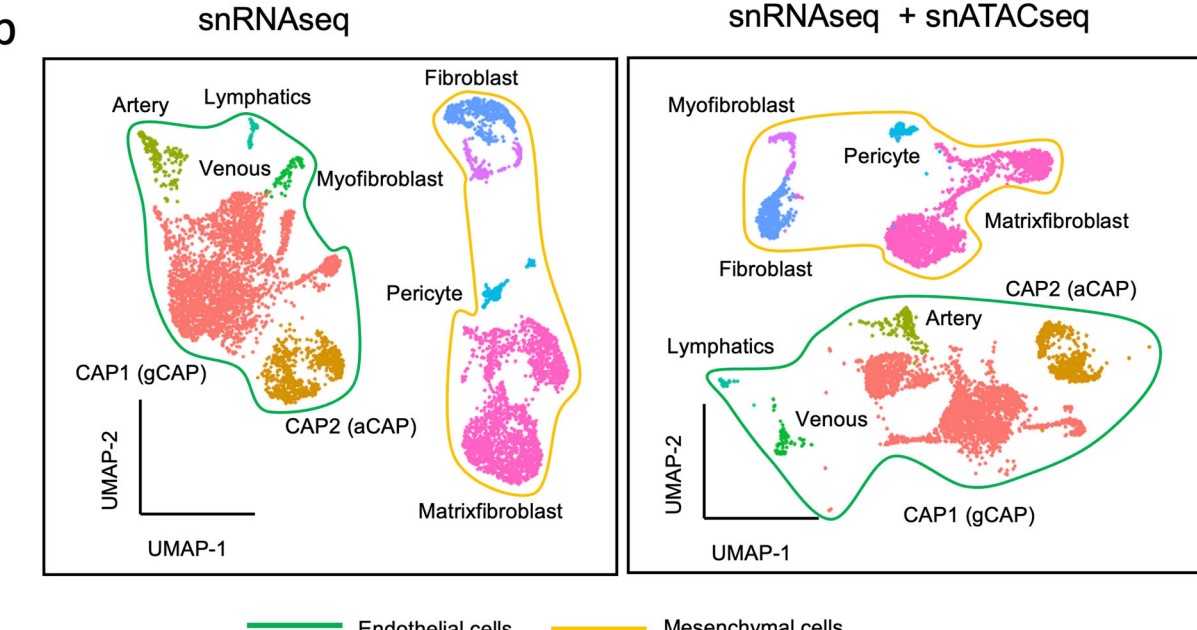

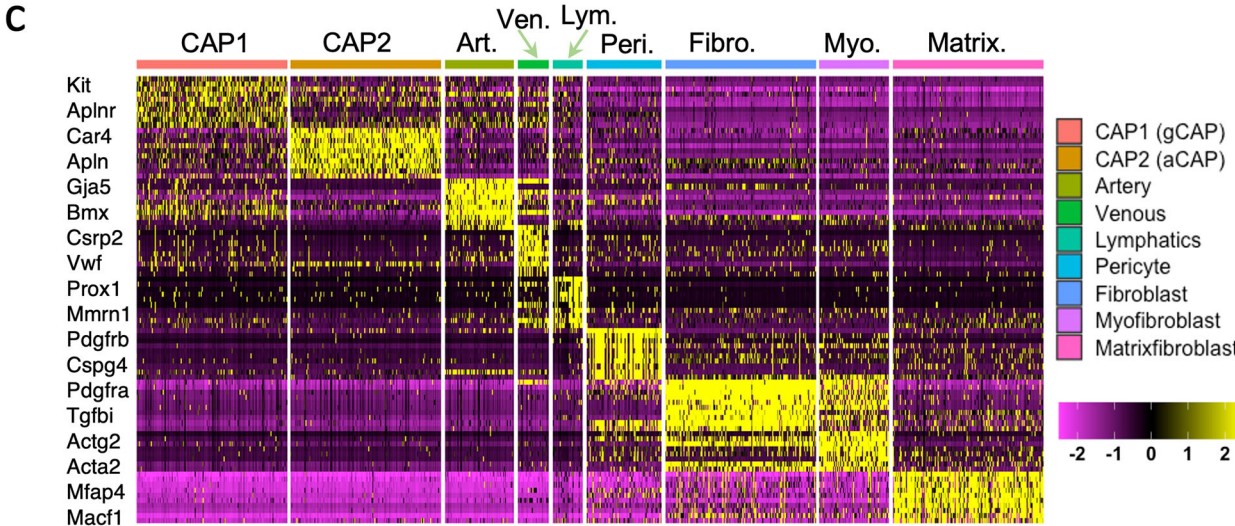

**Fig. 1 | Multiome (GEX + ATAC) profiling of the transcriptome and chromatin accessibility of *Foxf1*-expressing cells from lungs. a** The schematic diagram of multiome library preparation from *Foxf1*-expressing cells isolated from mouse E18.5 lungs. **b** Uniform Manifold Approximation and Projection (UMAP) plots show cell clustering based on snRNAseq alone or joint projection of snRNAseq and snATACseq. The *Foxf1*-expressing single nuclei were resolved into 9 cell clusters: pericyte (Peri), fibroblast (Fibro), and myofibroblast (Myo), matrixfibroblast (Matrix), CAP1 (gCAP), CAP2 (aCAP), Arterial (Art), Venous (Ven) and Lymphatics (Lym) endothelial cells. **c,** Heatmap shows the enriched genes in each cell cluster. Representative cell markers are listed for each cell cluster.

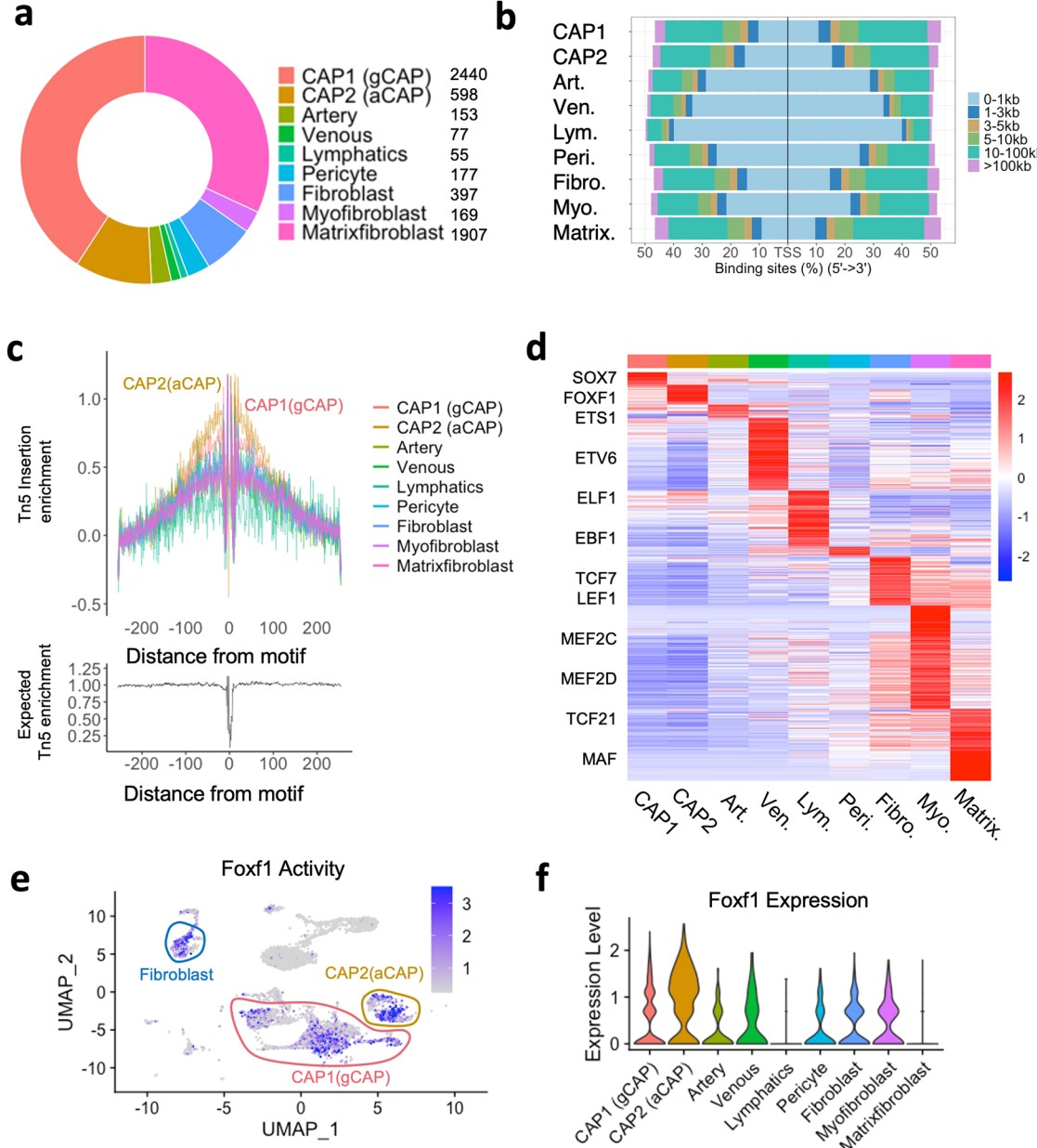

**Fig. 2 | The distribution of cell type-specific accessible chromatin regions in pulmonary cell clusters. a** The distribution of *Foxf1*-expressing nuclei across various cell types is shown by the donut plot. **b** The bar plot shows the statistical distribution of transcription factor binding sites relative to the transcription start site (TSS) for each cell cluster. **c** The transcriptional footprinting analysis shows that FOXF1-binding site are highly enriched in CAP1 (gCAP) and CAP2 (aCAP) cells. **d** The heatmap shows average chromVAR motif activity for each cell type. Transcription factors are indicated for each cell cluster. The color scale represents a z-score scaled in each row. **e** The featureplot shows that FOXF1 activity is enriched in fibroblast, pericytes, CAP1 and CAP2 endothelial cells. **f** Violin plots show the *Foxf1* expression in different cell clusters.

respectively (Fig. 2d). Interestingly, the highest activity of FOXF1 was predicated for CAP2, although FOXF1 activity and *Foxf1* mRNA were also detected in other cell types (Fig. 2c–f).

**Different accessibility of FEL enhancers in lung cell types**
To identify cell type-specific DNA regulatory regions in the *Foxf1* promoter, we compared promoter accessibility in each cell type. The *Foxf1* proximal promoter region near its transcriptional start site (TSS) was accessible in all cell clusters (Fig. 3a), predicting that this region is active in both endothelial and mesenchymal cell types. In contrast, distinct differentially accessible chromatin regions were identified, which varied among cell types (Fig. 3a). These four regions were named as FEL1, 2, 3 and 4 (*F*OXF1 *E*xpression in the *L*ung 1, 2, 3 and 4).

FEL1 and FEL3 regions were open in endothelial cells, whereas FEL2 and FEL4 were active in lung mesenchyme (Fig. 3a, b). When aligning our snATACseq data from E18.5 mouse lung with the publicly available snATACseq data from mouse E8.5 embryo, we found that the FEL2 enhancer is accessible in the embryonic mesenchyme (Supplementary Fig. 4), while other FEL enhancers are not open in the mesenchyme of E8.5 embryos. Alignment of our snATACseq data with publicly available bulk ATACseq data from various lung developmental stages, showed that the accessibility of endothelial FEL1 and FEL3 enhancers increases at E18.5, consistent with increased vascular development at canalicular-saccular stages of lung morphogenesis. In contrast, the accessibility of mesenchymal FEL2 and FEL4 enhancers was decreased by E18.5 (Supplementary Fig. 5). All FEL enhancers were evolutionarily

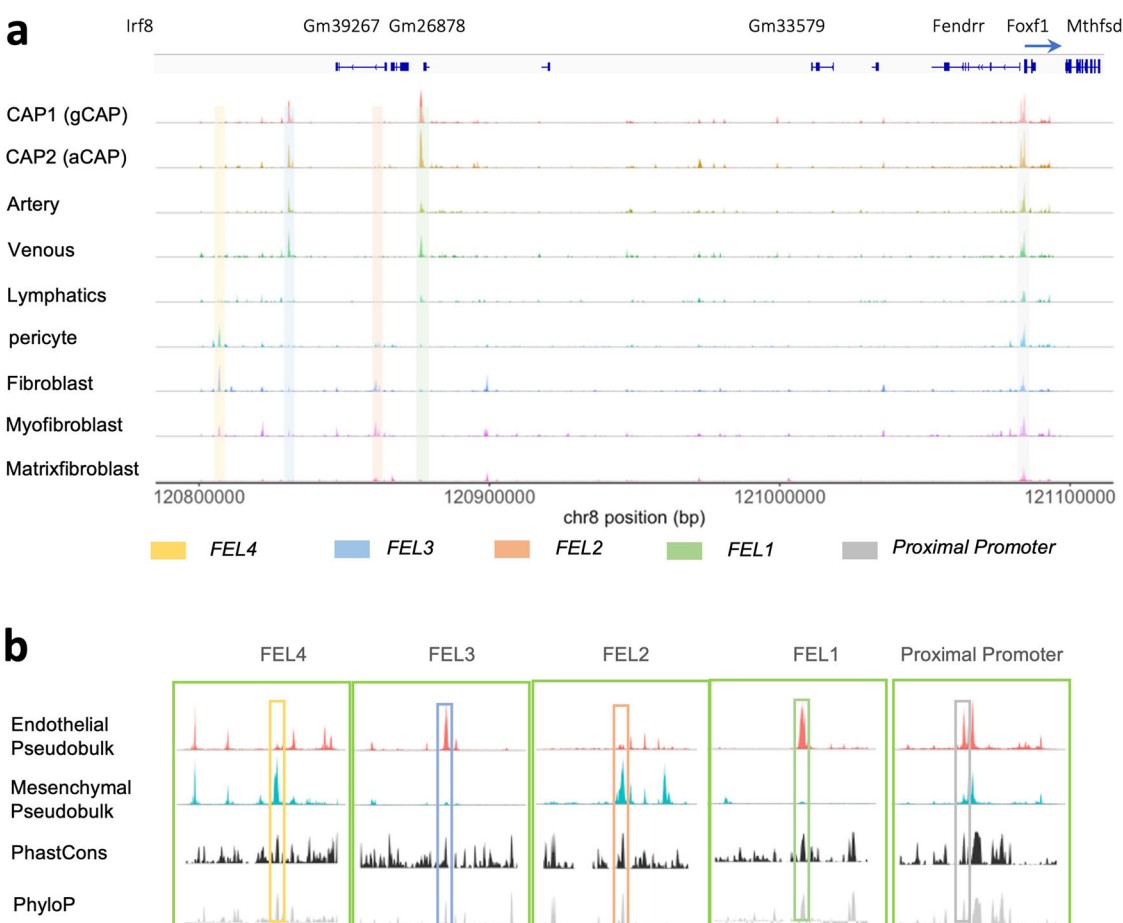

**Fig. 3 | *Foxf1* upstream regulatory enhancers exhibit differential accessibility in endothelial and mesenchymal cells. a** The coverage plot shows the differential accessibility of potential regulatory chromatin in the *Foxf1* upstream region. TSS: transcription start site. The *Foxf1* proximal promoter region is accessible in all Foxf1-expressing cell types. There are four accessible regions in the *Foxf1* upstream regulatory sequences: FEL1, FEL2, FEL3, and FEL, and the four upstream regions are differentially accessible among Foxf1-expressing cell clusters. **b** The alignment of pseudo-bulk endothelial and mesenchymal accessibility shows that FEL1 and FEL3 are exclusively accessible in endothelial cells, whereas FEL2 and FEL4 are accessible only in mesenchymal cells. The alignment to phastCons and phyloP evolutionary conservation tracks shows that the accessible peaks are evolutionarily conserved across placental mammals.

conserved among mammals (Supplementary Figs. 6, 7). Thus, FEL distal regulatory elements in the *Foxf1* promoter are differentially accessible in lung endothelial vs mesenchymal cells.

## FOXF1 activates endothelial enhancers FEL1 and FEL3

Since FOXF1 protein binds to and activates the proximal *Foxf1* promoter in an autoregulatory loop[9], we examined if FOXF1 also regulates distal FEL enhancers. We aligned our snATACseq dataset with published FOXF1 ChIPseq data[27] to identify FOXF1-binding sites in the open chromatin of the putative FEL enhancers. FOXF1 binds to FEL1 and FEL3 endothelial-specific enhancers but not to FEL2 and FEL4 mesenchymal enhancers (Fig. 4a), suggesting an endothelial-specific autoregulation. Consistent with the importance of the FOX:ETS motifs for endothelial gene regulation[28], evolutionarily conserved FOX:ETS DNA binding motifs were found in both FEL1 and FEL3 endothelial enhancers, as revealed by the joint analyses of ChIPseq and snATACseq data (Fig. 4b and Supplementary Fig. 6). To test if the FOXF1-binding sites in FEL1 and FEL3 were functional, we cloned the FEL1 and FEL3 enhancers into Luciferase (Luc) reporter plasmids. Co-transfection of these Luc reporter plasmids with CMV-FOXF1 expression vector revealed that FOXF1 induced the transcriptional activity of both FEL1 and FEL3 (Fig. 4c, d). Site-directed mutagenesis of the FOXF1-binding sites decreased the activity of FEL1 and FEL3 (Fig. 4c, d), demonstrating that these FOXF1-binding sites are likely to be autoregulated. Interestingly,

FOXF-binding sites in both FEL1 and FEL3 were located near conserved ETS binding motifs (Fig. 4b), consistent with published studies demonstrating the cooperation of FOX and ETS sites in endothelial regulatory elements[28]. Simultaneous disruption of both FOX and ETS sites was carried out in FEL1 and FEL3 enhancers using site-directed mutagenesis (Supplementary Fig. 8a). In co-transfection experiments, FOXF1 cooperated with ETS transcription factors FLI1 and ERG to stimulate transcriptional activity of FEL1 and FEL3 enhancers (Supplementary Fig. 8b, c), consistent with the FOX:ETS synergy in the proximal *Foxf1* promoter region[9]. Since most enhancers are bound by the transcriptional coactivator EP300[29], we aligned our snATACseq data to EP300 Chipseq datasets from both mouse fetal and adult lung[29,30]. EP300 specifically bound to the endothelial-specific FEL1 and FEL3 enhancers, but not to the proximal *Foxf1* promoter region (Fig. 4e). Taken together, FOXF1 binds to and transcriptionally activates its own endothelial-specific FEL1 and FEL3 enhancers.

## FEL2 and FEL4 enhancers are regulated by EBF1 and GLI1

Next, we examined the transcriptional networks regulating the activity of FEL2 and FEL4 mesenchymal enhancers. FEL4 was accessible in both fibroblasts and pericytes (Fig. 5a and Supplementary Fig. 7a). We use CisBP motif library[31] to identify transcription factor binding sites. Forkhead binding motif was not detected in FEL2 and FEL4 based on joint analysis of ChIPseq and snATACseq data, suggesting the

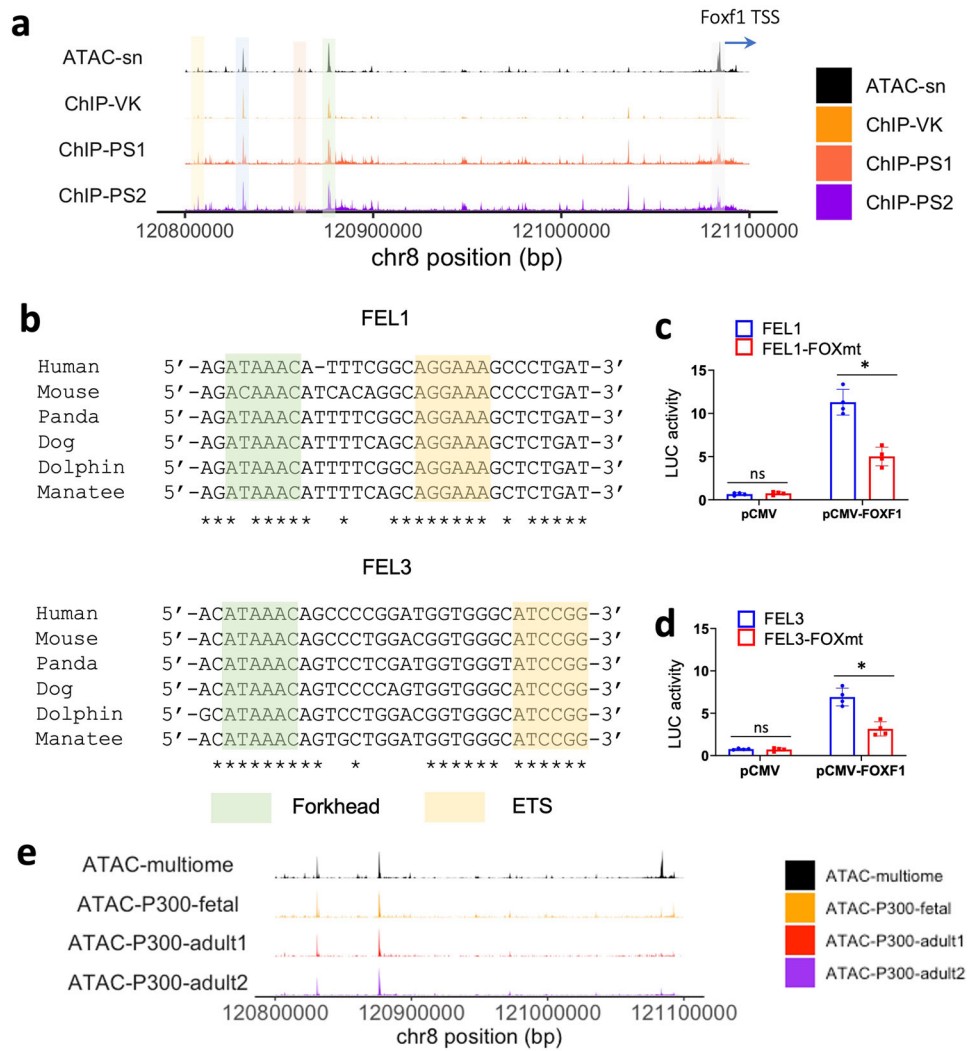

**Fig. 4 | Foxf1 autoregulation in pulmonary endothelial cells. a** The alignment of snATACseq data from single nuclei multiome to the FOXF1 ChIPseq data shows that FOXF1 binds to the *Foxf1* proximal promoter region as well as FEL1 and FEL3 in endothelial cells. **b** Sequences show Forkhead-binding motifs and ETS-binding motifs in FEL1 and FEL3 enhancers of six evolutionarily divergent mammals. **c, d** Disruption of Fox-binding motif in FEL1 and FEL3 significantly reduces the activity of these enhancers after transfection with CMV-*Foxf1* plasmid, as revealed by dual luciferase assay. Data are presented as mean ± SD (**c, d**). $n = 4$ biological replicates for (**c, d**). $p < 0.01$ is **$p < 0.05$ is *, ns is not significant. Two-sided Student's *t* test is used for statistical analysis. The pvalue is 0.01645 (**c**) and 0.01933 (**d**). **e** The alignment of EP300 Chipseq datasets in fetal and adult lung shows that FOXF1 and P300 bind to the same endothelial specific regulatory elements in distal but not in proximal *Foxf1* region. Source data are provided as a Source Data file.

mesenchymal *Foxf1* enhancers are not subject to autoregulation. However, an evolutionarily conserved binding motif for EBF (Early B-cell factor) family transcription factor was identified in FEL4. Single-cell RNA sequencing[9] revealed that *Ebf1* mRNA is enriched in lung pericytes, whereas the *Ebf2*, *Ebf3* and *Ebf4* mRNAs were undetectable (Fig. 5b). The correlation analysis revealed that *Foxf1* and *Ebf1* are co-expressed in a subset of pericytes (Fig. 5c). To determine the role of EBF1 in transcriptional regulation of the FEL4 enhancer, we cloned the FEL4 region into the Luc reporter plasmid. Co-transfection of the FEL4-Luc reporter plasmid with CMV-EBF1 expression plasmid demonstrated that EBF1 stimulates the FEL4 enhancer activity (Fig. 5d). Site-directed mutagenesis of the EBF1-binding site in FEL4 enhancer completely inhibited the activation of FEL4 by CMV-EBF1 (Fig. 5d), demonstrating that EBF1 stimulates FEL4 via the EBF-binding motif. However, a 50% decrease of *Ebf1* expression by siRNA in MFLM-91U cells was insufficient to decrease *Foxf1* mRNA or significantly inhibit FEL4 enhancer activity (Supplementary Fig. 9a, b). Interestingly, the early embryonic FEL2 enhancer contains an evolutionarily conserved GLI-binding site (Supplementary Fig. 7b), a finding consistent with

previous studies demonstrating that SHH/GLI is required for *Foxf1* gene expression in the early embryonic mesenchyme[32,33]. Site-directed mutagenesis of the GLI site in FEL2-Luc reporter plasmid demonstrated that FEL2 is activated by GLI1 (Supplementary Fig. 9c, d). In contrast, FEL4 enhancer was not activated by GLI1 (Supplementary Fig. 9d). Thus, FEL2 and FEL4 mesenchymal enhancers are regulated by GLI1 and EBF1, respectively.

**Multiome analysis of lungs with non-coding FOXF1 deletions**

Genomic deletions and point mutations in *FOXF1* gene locus located in 16q24 are linked to human ACDMPV[8]. We analyzed 21 cases of ACDMPV with previously mapped genome deletions[8,12] and found that *FOXF1* coding sequences were intact in 12 cases (57%) (Supplementary Fig. 10a). Among these 12 ACDMPV cases, 11 cases (92%) had deletions that include one or more FEL regulatory elements (Supplementary Table 1). To determine if the upstream FEL1-4 enhancers contribute to pathogenesis of ACDMPV, we examined the multiome (snRNA + snATAC) datasets derived from lungs of two ACDMPV patients with non-coding *FOXF1* deletion (designated as P1 and P2,

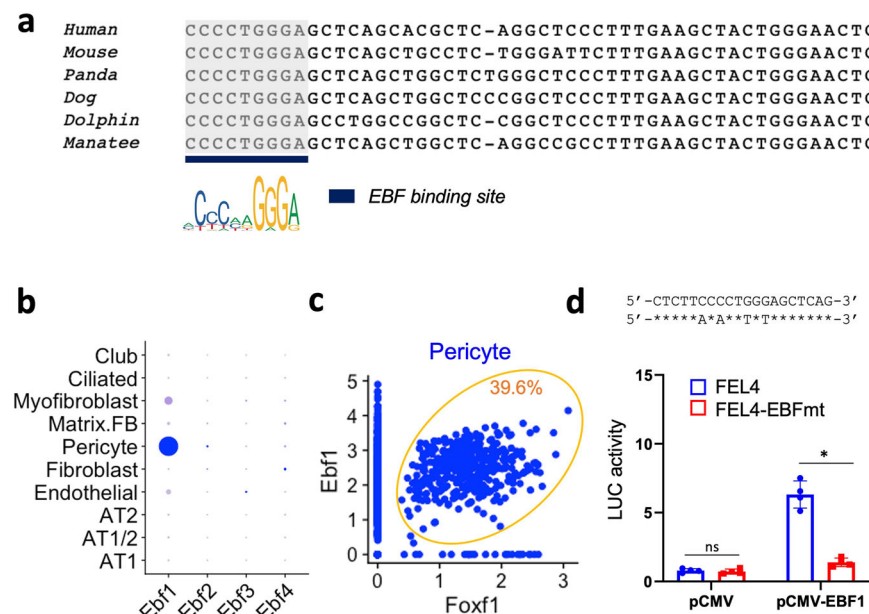

**Fig. 5 | Foxf1 expression in pericytes is regulated by EBF1. a** The sequence alignment of the evolutionarily conserved FEL4 enhancers shows the EBF binding site. The EBF-binding motif is shown in the EBF-binding site located in the FEL4 enhancer. **b** EBF1 is the most highly expressed member of the Ebf family of transcription factors in E18.5 lung as shown by single cell RNA sequencing. **c** The dot plot shows a correlation analysis between *Foxf1* and *Ebf1* in pulmonary pericytes from E18.5 lungs. The percentage of Foxf1+Ebf1+ pericytes is 39.6%. **d** Disruption of the EBF1 binding motif significantly reduces the activity of the FEL4 enhancer after co-transfection with CMV-*Ebf1* plasmid, in the dual luciferase assay, and $n = 4$ biological replicates. Data are presented as mean ± SD. $p < 0.01$ is **$p < 0.05$ is *, ns is not significant. Two-sided Student's $t$ test is used for statistical analysis, and $p$-value is 0.01096. Source data are provided as a Source Data file.

respectively)[34]. P2 patient exhibited a deletion of FEL2, 3, and 4, but not FEL1. The P2 genomic deletion had a ~6 kb overlap with the previously characterized 75 kb SDR (shared deletion region)[12]. P1 patient exhibited a deletion of all four FEL enhancers. The P1 genomic deletion covered both the 75 kb SDR and 60 kb SRO regions (Supplementary Fig. 10b, c) that were previously implicated in pathogenesis of ACDMPV[12].

Pulmonary cells from both ACDMPV patients aligned well based on unsupervised clustering and UMAP embedding (Supplementary Fig. 11a). Both P1 and P2 genomic deletions were located upstream of the coding *FOXF1* gene region (Fig. 6a, b and Supplementary Fig. 11b). To examine chromatin accessibility in FEL enhancers, the cells were subdivided into three main groups based on developmental cell origins: endothelial cells, mesenchymal cells, and epithelial cells. (Fig. 6c and Supplementary Fig. 11c). The chromatin accessibility of cell-specific FEL enhancers was compromised in ACDMPV patients (Fig. 6d and Supplementary Fig. 11d), compared with the published snATACseq data from healthy donor lungs[21] (Fig. 6e). Next, the snRNAseq dataset from patient P2 was processed using unsupervised nonlinear clustering and UMAP embedding (Supplementary Fig. 12a). Based on cell-specific markers, CAP1 and CAP2 capillary endothelial cells were rare in ACDMPV lung (Supplementary Fig. 12b), a finding consistent with the histological features in ACDMPV patients[35]. The FEL1 enhancer was not accessible in endothelial cells from P2 patient (Supplementary Fig. 12c). In contrast, FEL1 was accessible in endothelial cells from healthy human donor lungs (Supplementary Fig. 13a, b). Interestingly, human donor cells showed accessibility of FEL2 in both endothelial cells and fibroblasts (Supplementary Fig. 13b), and the functional GLI-binding site in FEL2 enhancer was present in the region of increased chromatin accessibility for both endothelium and fibroblasts (Supplementary Fig. 13c). The overall genomic organization flanking the *FOXF1* gene was evolutionarily conserved as shown by comparative analysis of human and mouse single-nuclei ATACseq datasets based on synteny between human chromosome 16 and mouse

chromosome 8 in the region between *GINS2* and *BANP* genes (Supplementary Fig. 14a–e).

## Functional validation of FEL1 and FEL4 enhancers in vivo

To determine whether the distal regulatory elements are required for cell differentiation during lung development, we selected one endothelial (FEL1) and one mesenchymal (FEL4) enhancer for functional validation using a recently generated mouse embryonic stem cell line (ESC) containing the *Foxf1-GFP* knock-in reporter and a constitutively active *CAG-tdTomato* transgene[15]. The Cpf1 (Cas12a)-based multiplex CRISPR (Clustered Regularly Interspaced Short Palindromic Repeats) genome-editing technique was used to disrupt either FEL1 or FEL4 in *Foxf1-GFP; CAG-tdTomato* ESCs (Fig. 7a). Subsequently FEL1$^{-/-}$ and FEL4$^{-/-}$ ESC clones were established (Supplementary Figs. 15a, b and 16a, b), and the deletion of either 1810bp (flanking FEL1) or 1140 bp (flanking FEL4) DNA sequences were verified by Sanger sequencing (Supplementary Figs. 15c, d and 16c, d).

We injected mutant ESCs into blastocysts of wild-type (WT) mice and implanted the ESC-complemented embryos into surrogate females (Fig. 7b). Mutant embryos were allowed to develop until E18.5 and then harvested. Control embryos were produced via blastocyst complementation of parental *WT* ESCs (without FEL1 or FEL4 deletion). Knockout of either FEL1 or FEL4 in ESCs did not reduce the efficiency of blastocyst complementation, as demonstrated by equal contribution of control and mutant cells into chimeric lungs (Supplementary Fig. 17a, b). To examine the efficiency of differentiation of mutant ESCs into different lung cell types, we used flow cytometry for CD31, CD45, CD140a, CD326, and CD146 cell surface markers to identify pericytes, fibroblasts, epithelial cells, and endothelial cells that were derived from donor ESCs in chimeric lungs (Fig. 7c). Deletion of FEL1, but not FEL4, reduced the numbers of endothelial cells derived from ESCs (Fig. 7d, e). Thus, FEL1 is required for the differentiation of pluripotent ESCs into lung endothelial cells in vivo, whereas FEL4 is dispensable for endothelial differentiation. Consistent with the mesenchymal specificity of the

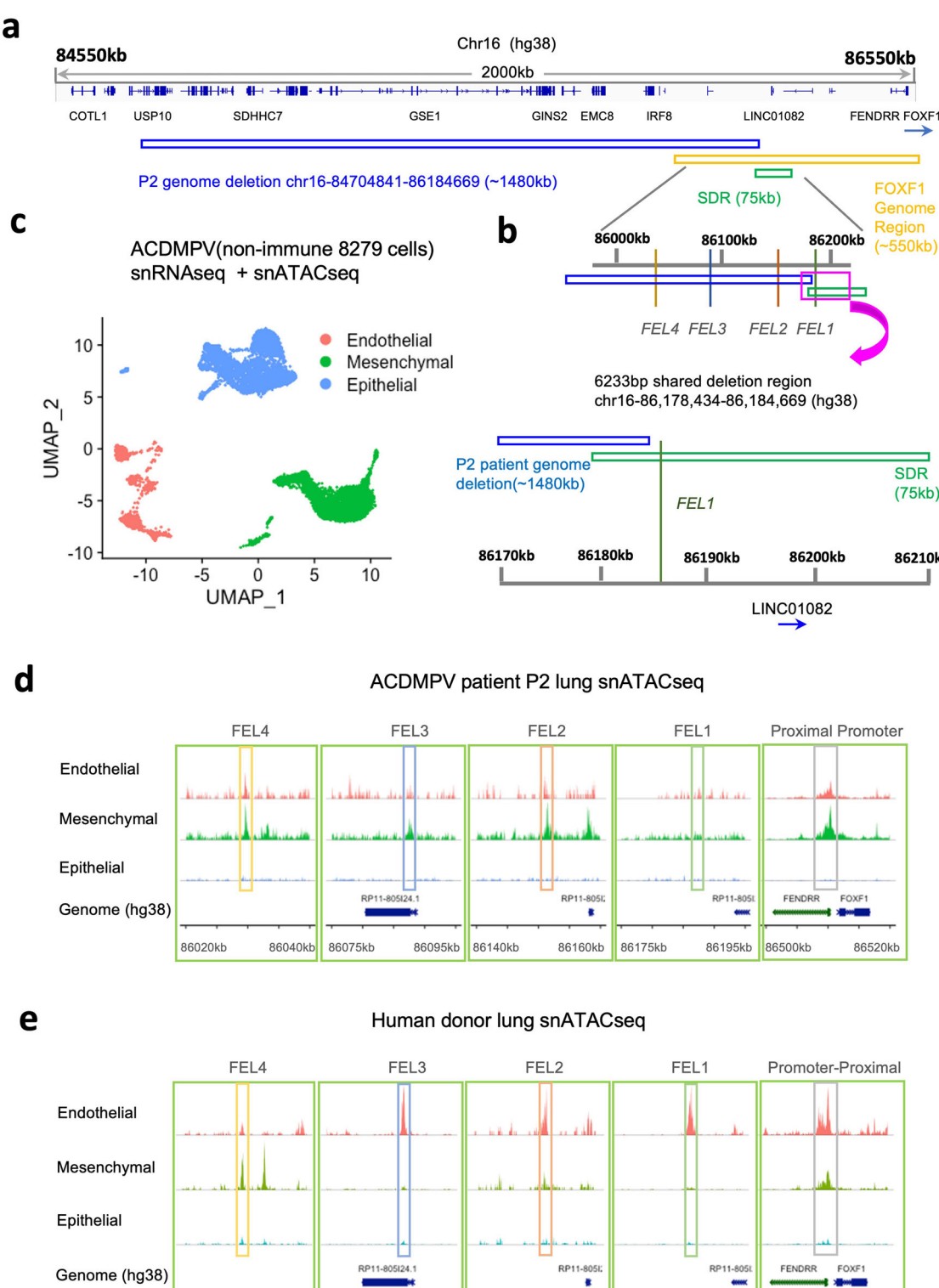

**Fig. 6 | Genomic regulation of *FOXF1* in ACDMPV patient lung. a** Schematic diagram shows the genomic deletion (blue) in P2 ACDMPV patient in the context of 16q24.1 region. The *FOXF1* genomic region is highlighted by orange box. The green box indicates the 75 kb shared deletion region (SDR). **b** The illustration shows the partial overlap of the *FOXF1* genomic deletion in patient P2 which includs FEL2-4, but not FEL1. The *FOXF1* genome deletion boundary is very close (<1 kb) to FEL1. The annotated coordinates are based on the human genome assembly hg38. **c** The UMAP joint projection and clustering of non-hematopoietic lung cells

(corresponding to CD45-). **d** The alignment of five *FOXF1* regulatory elements is performed using single nuclei multiome sequencing from patient P2. The accessibility of the *FOXF1* proximal promoter region is detected, but the accessibility of endothelial regulatory elements is significantly compromised. **e** The alignment of five *FOXF1* regulatory elements is performed using single nuclei multiome sequencing from donor lung. The 20 kb adjacent regions are shown for each regulatory element.

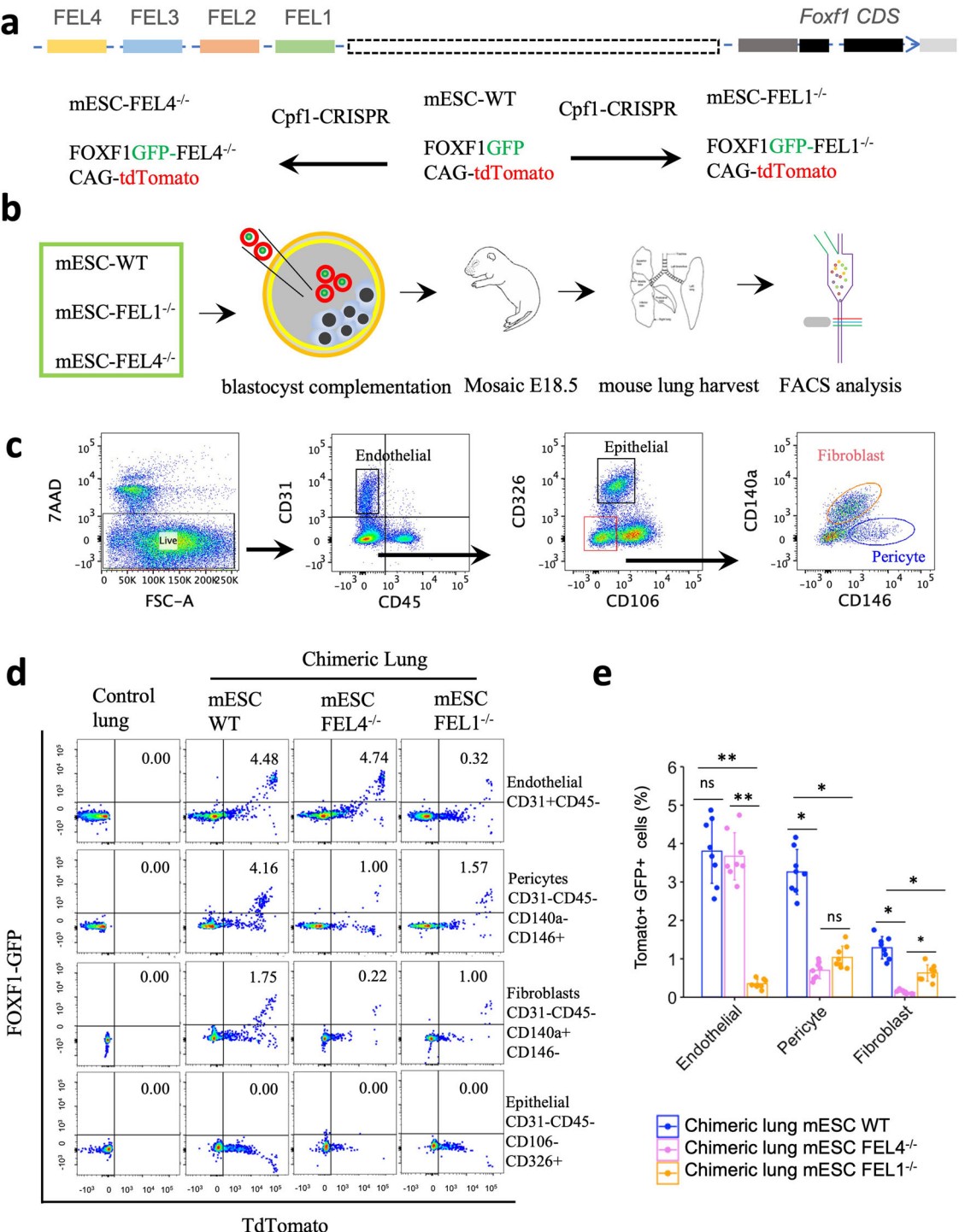

**Fig. 7 | Functional assessment of FEL1 and FEL4 enhancers using blastocyst complementation. a** The schematic diagram shows both distal and proximal elements regulating FOXF1 expression in the lung. The Cpf1-based CRISPR duplex genome editing was used to disrupt FEL1 and FEL4 enhancer in mouse embryonic stem cell (mESCs). **b** The schematic illustration shows the analytic pipeline that uses blastocyst complementation and flow cytometry. **c** The flow cytometry gating strategy shows the identification of pulmonary pericytes, fibroblasts, epithelial cells and endothelial cells based on cell surface markers. **d** FACS analysis of the in vivo cell differentiation of mESCs donor into endothelial cells, epithelial cell, pericytes and fibroblasts by blastocyst complementation. **e** The quantification of

cells derived from donor tdTomato+GFP+ ESCs based on cell-specific markers. Disruption of FEL1 decreases differentiation of donor mESCs into endothelial cell lineage, and mESC differentiation into pulmonary fibroblasts and pericytes is decreased after disruption of either FEL1 or FEL4 in mESCs, and $n = 8$ biological replicates. Data are presented as mean ± SD. Two-sided Student's $t$ test is used for statistical analysis. $p < 0.01$ is **$p < 0.05$ is *, ns is not significant. The $p$-values for endothelial: 0.13879, 0.00547, 0.00571; The pvalues for pericyte: 0.02852, 0.01504, 0.19342; The pvalues for fibroblast: 0.04024, 0.04331, 0.0393. Source data are provided as a Source Data file.

FEL4 enhancer, knockout of FEL4 decreased the numbers of pericytes and fibroblasts derived from ESCs (Fig. 7d, e). There were no *Foxf1*-expressing epithelial cells among donor cells (Fig. 7d), a finding consistent with published studies demonstrating that *Foxf1* expression is restricted to mesoderm-derived cells during lung development[36]. Interestingly, genomic disruption of FEL1 also affected the numbers of pericytes and fibroblasts (Fig. 7e), suggesting that FEL1 influences the differentiation of pluripotent ESCs into lung pericytes and fibroblasts in addition to endothelial differentiation.

To determine whether deletion of FEL1 or FEL4 enhancers affects endogenous mouse *Foxf1* mRNA in relevant cell types, we used blastocyst complementation followed by single-cell RNAseq analysis of FACS-sorted tdTomato+ lung cells derived from mutant or WT mouse ESCs in chimeras. The cells were integrated from three single-cell RNAseq datasets: WT (control), FEL1$^{-/-}$ (MT1) and FEL4$^{-/-}$ (MT4). Six main cell clusters were identified based on gene expression signatures: Endothelial, Epithelial, Fibroblast, Myofibroblast, Matrixfibroblast and Pericyte (Fig. 8a, b). All cell clusters were present in three individual single-cell RNAseq datasets (Fig. 8c). *Foxf1* mRNA was undetectable in FEL1$^{-/-}$ endothelial cells (Fig. 8d). In contrast, homozygous FEL4 deletion selectively abolished endogenous *Foxf1* mRNA in fibroblasts and pericytes (Fig. 8d). The percentage of ESC-derived endothelial cells was reduced after deletion of FEL1 but not FEL4 (Supplementary Fig. 18a, b), a finding consistent with flow cytometry data (Fig. 7d, e). While the percentage of fibroblasts was reduced (Supplementary Fig. 18a, b), the ratio of myofibroblasts was higher after deletion of FEL1 (Supplementary Fig. 18c, d). Based on smooth muscle cell markers[37], we further distinguished smooth muscle cells from myofibroblasts (Supplementary Fig. 19a, b). The cell sub-clustering demonstrated that ESC-derived FEL1$^{-/-}$ cells exhibit a higher ratio of myofibroblasts, whereas FEL4$^{-/-}$ cells exhibit a higher ratio of smooth muscle cells compared to WT control (Supplementary Fig. 19c–e). Taken together, FEL1 and FEL4 distal enhancers are cell-specific regulators of endogenous *Foxf1* gene expression and play important roles in the differentiation of pulmonary endothelial and mesenchymal cells from pluripotent ESCs in vivo.

## Discussion

The *FOXF1* gene is located within a protein-coding desert region spanning ~300 kb (mouse) or ~500 kb (human), and it is regulated by promoter and enhancer sequences[38]. Published human genetic studies demonstrated that the loss of the 257 kb region located upstream of *FOXF1* coding sequences is associated with ACDMPV[8,39,40]. FOXF1 adjacent non-coding developmental regulatory RNA (*FENDRR*) is transcribed in the opposite direction to the neighboring *FOXF1* gene with which it shares promoter[41]. The lung-specific distal enhancer region, located ~270 kb upstream to the *FOXF1* gene, was found to interact with the *FOXF1* promoter region to regulate transcription of both *FOXF1* and *FENDRR*[13,42,43]. Chromosome conformation capture (3C) assay in a mouse model identified two long-range enhancers located 200 kb upstream of the mouse *Foxf1* gene coding region[44]. While *FOXF1* expression is subject to cis regulation from upstream regulatory elements of bimodal structure[45,46], the molecular mechanisms regulating the cell specificity of *FOXF1* enhancers remain largely unknown. An important contribution of the present study is that *FOXF1* is regulated by evolutionarily conserved endothelial and mesenchymal FEL enhancers, the loss of which can contribute to pathogenesis of human ACDMPV.

Composite FOX:ETS binding sites are present in many known endothelial-specific enhancers in both mouse and zebrafish[28]. FOXF1 synergized with ETS transcription factor FLI1 to activate the proximal murine *Foxf1* promoter[9]. In the present study, we found evolutionarily conserved FOX:ETS motifs in distal endothelial-specific FEL1 and FEL3 enhancers, whereas these motifs were absent in

mesenchymal FEL2 and FEL4 enhancers. These data are consistent with the importance of FOX:ETS composite motifs in endothelial-specific gene expression. Since the disruption of the FOX sites within FOX:ETS motifs decreased the ability of the FOXF1 expression vector to activate endothelial FEL1 and FEL3 enhancers, our results indicate that FOXF1 regulates its own gene expression, at least in part, through FEL1 and FEL3.

The Hedgehog (HH)/GLI signaling pathway is one of the major regulators of lung development, and FOXF1 is a known downstream target of HH/GLI signaling in lung morphogenesis[32,47]. Consistent with previous studies, we identified an evolutionarily conserved GLI-binding motif in the FEL2 enhancer, disruption of which prevented the activation of FEL2 by GLI1. Interestingly, a second distinct FOXF1 mesenchyme-specific enhancer (FEL4) is regulated by EBF1 transcription factor, as evidenced by the loss of the FEL4 enhancer activity after disruption of the EBF1-binding site. EBF1 was recently identified as an important transcription factor in pulmonary pericytes[48]. EBF1 was also shown to be critical for B-lymphocyte differentiation[49] and during lung repair after LPS-mediated injury[50]. Our findings indicate that FEL2 and FEL4 mesenchymal enhancers in *FOXF1* promoter are differentially regulated by GLI and EBF1 during lung development.

Multiome sequencing is a powerful technique to identify regulatory elements critical for gene expression. However, the methods of experimental validation of these regulatory elements are not well developed. In the present study, we used the multiome technique to systematically profile the chromatin accessibility of single cells and to identify the evolutionarily conserved upstream regulatory elements in the *FOXF1* gene locus. To functionally validate the *FOXF1* regulatory elements identified by multiome, we took advantage of blastocyst complementation to monitor the effect of upstream regulatory elements on cell differentiation in vivo[51]. The Cpf1-dependent CRISPR multiplex genome editing provides an excellent platform for specific knockout analysis of non-coding regulatory sequences due to its AT-rich protospacer adjacent motif (PAM) and short guide RNA[52]. Single-cell RNA sequencing of donor cells derived from mutant ESCs via blastocyst complementation further supports the cell specificity of distal FEL enhancers and their involvement in the regulation of *FOXF1* gene expression. While the combined use of multiome and blastocyst complementation of CRISPR/Cpf1-edited ESCs provides an efficient in vivo pipeline to evaluate the number of ESC-derived cells and their gene expression patterns, this analysis has important limitations related to the evaluation of requirements for FEL enhancers during embryonic development. Mouse genetic models with global or conditional inactivation of individual FEL enhancers are needed to investigate requirements for FEL enhancers in vivo.

We have performed a comprehensive multiome analysis of chromatin accessibility in the *FOXF1* gene locus in mouse and human lungs. FOXF1 expression is regulated by its own proximal promoter and distal FEL enhancers that selectively mediate either endothelial-specific or mesenchyme-specific FOXF1 expression (Supplementary Fig. 20). The endothelial FOXF1 expression is dependent on FOXF1:ETS motifs in FEL1 and FEL3 enhancers, whereas the FEL4 enhancer is regulated by EBF1 in mesenchymal cell lineages, such as fibroblasts and pericytes. The FEL2 enhancer is regulated by GLI1 in the early lung mesenchyme (Supplementary Fig. 20). Pulmonary mesenchymal and endothelial cells share common mesodermal progenitors that originate from lateral plate mesoderm[53–55]. It is possible that FEL2 is a developmentally regulated *FOXF1* enhancer that plays a role in mesodermal progenitor cells. However, the chromatin accessibility of FEL2 is different in mouse and human lung cells, likely reflecting interspecies genetic or epigenetic differences. Based on single nuclei ATAC sequencing data from human lungs, it is possible that FEL2 can be active in both endothelial and

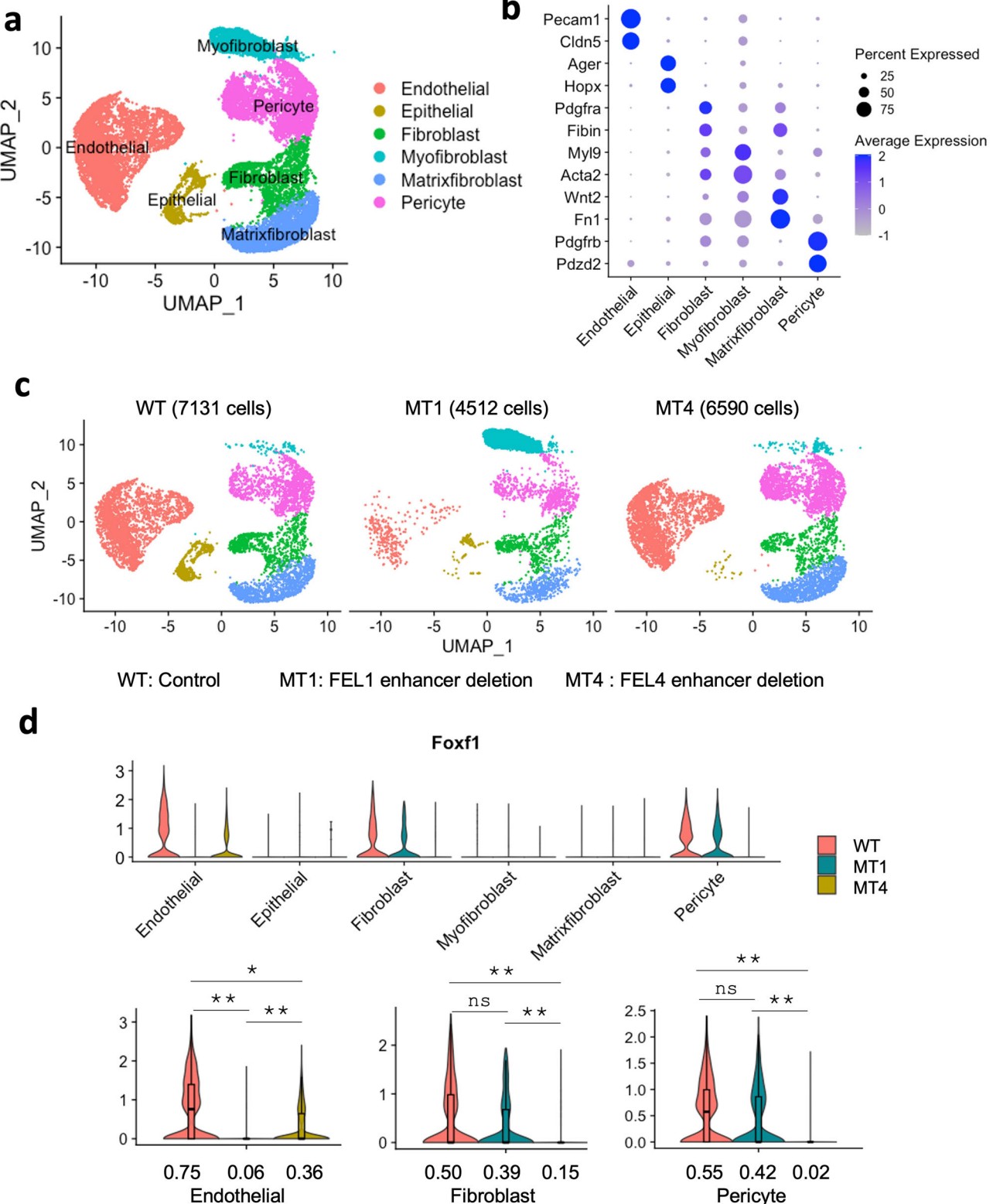

mesenchymal cell lineages depending on the timing of lung development or duration of therapeutic interventions in ACDMPV. Since the disruption of FEL enhancers is common for ACDMPV, the detailed characterization of the *FOXF1* regulatory elements will provide further insights into molecular mechanisms critical for ACDMPV pathogenesis and can improve genetic diagnosis of this devastating congenital disorder.

## Methods

### Ethics statement

The data, analytic methods, and study materials will be made available upon request from the corresponding author of this manuscript. All animal studies were approved by the Institutional Animal Care and Use Committee of the Cincinnati Children's Hospital. All mice were kept under SPF (specific-pathogen free) conditions in 12/12 light/dark cycle,

**Fig. 8 | Single-cell RNA sequencing analysis shows changes of endogenous *Foxf1* mRNA in FEL1⁻/⁻ and FEL4⁻/⁻ lung cells derived via blastocyst complementation.**
**a** The integrative single cell RNAseq analysis of tdTomato+ chimeric lung cells derived from control mESC s(WT), FEL1⁻/⁻ mESCs (MT1), and FEL4⁻/⁻ mESCs (MT4) via blastocyst complementation. The cells from the three datasets were grouped into six cell clusters based on mRNA signatures: Endothelial, Epithelial, Fibroblast, Myofibroblast, Matrixfibroblast and Pericyte. **b** DotPlot shows mRNAs used as representative markers for each cell cluster. **c** The unsupervised clustering and UMAP projection of single cells from each individual WT (7131 cells), MT1 (4512 cells) and MT4 (6590 cells) datasets are shown. **d** VlnPlot analysis indicates that *Foxf1* mRNA is decreased in MT1 endothelial cells as well as in MT4 pericytes and fibroblasts (upper panel). The box in the box plot represents the middle 50% of the data, with the lower and upper edges of the box indicating the first and third

quartiles, respectively. The line inside the box represents the median. The whiskers extend from the box to the minimum and maximum values. The average *Foxf1* expression in each cell type is indicated in bottom panels for endothelial cells (WT:2722), (MT1:296) and (MT4:2543); fibroblast (WT:978), (MT1:615) and (MT4:646); pericyte (WT:1363), (MT1:783) and (MT4:1937). The numbers following each genotype indicated the cells used to calculate Foxf1 expression level in each genotype from single cell RNA sequencing data. One-way analysis of variance (ANOVA) analysis was used to determine statistical significance. Multiple means were compared using one-way ANOVA with the post-hoc Tukey's test. The pvalues for endothelial with significant difference: 0.00535, 0.00856, 0.01851; The pvalues for pericyte with significant difference: 0.00643, 0.00518; The pvalues for fibroblast with significant difference: 0.00406, 0.00313. $p < 0.01$ is **$p < 0.05$ is *, ns is not significant.

18–23 °C and 40–60% humidity. The Foxf1-GFP mouse line was generated and maintained in C57BL/6 genetic background with CRISPR technology. We examined the animals daily to ensure good health and well-being by monitoring for any signs of pain or distress. Mice were euthanized by intraperitoneal injection of pentobarbital sodium and embryos were collected at E18.5 for downstream analysis. The NIH guidelines for laboratory animal care were strictly followed.

### Single-nuclei 10X multiome library preparation and data analysis of murine cells

The embryonic lungs were pooled from six littermate embryos for FACS sorting and single-nuclei multiome analysis. Sex of embryos was not determined prior to combining the samples. For 10X single-nuclei multiome ATAC + GEX sequencing, single-cell suspensions were prepared by dissociation of lungs from *Foxf1-GFP* transgenic mice using 0.2 mg/ml Liberase TM (Roche) and 100U/ml Deoxyribonuclease I (DNase I, Sigma) as described[56]. The single cell suspension was FACS-sorted for GFP signal to enrich FOXF1-expressing cells. Sorted cells were lysed and nuclei were isolated according to the manufacturer's recommendations described in Nuclei Isolation for Single Cell Multiome ATAC + Gene Expression Sequencing protocol CG000365 (10X genomics). Single-nuclei libraries were prepared in the Single Cell Genomics Core Facility of Cincinnati Children's Hospital Medical Center using Chromium Next GEM Single Cell Multiome ATAC + Gene Expression Reagent bundle (PN-1000283), Chromium Next GEM chip single cell kit (PN-1000234) and Chromium Next GEM Single Cell multiome ATAC + Gene Expression User Guide (CG000338). Briefly, single nuclei were transposed in a thermal cycler and the suspension containing transposed nuclei was then loaded into a microfluidic chip in chromium controller to produce gel beads-in-emulsion (GEMs). The libraries were sequenced using NovaSeq 6000 (Illumina). The RNA and ATAC reads from multiome single-nuclei libraries were demultiplexed and processed using the Cell Ranger ARC software package (10x Genomics) and mouse reference mm10. Nuclei that passed Cell Ranger ARC filters were used as input to R package Signac[17]. Quality controls for RNA library included RNA filters that were less than 1000 and more than 40000 counts and higher than 10% mitochondrial reads. Quality controls for the ATAC library included cells containing between 1000 and 100,000 UMIs, <1 nucleosome signal values and ≥1 transcription start site (TSS) enrichment score. Scrublet was applied to the RNA library to identify and remove potential doublets. After quality controls, 5973 nuclei were used for downstream analysis using R packages Seurat (version 4.0) and Signac (version 1.7). Cell types were identified based on the RNA library alone, and the cell types were further validated using markers from the online Lunggens portal https://research.cchmc.org/pbge/lunggens/. The RNA library down-dimension analysis was performed from RunUMAP implemented in R package Seurat 4.0[57]. The joint neighbor graph analysis for combined RNAseq and ATACseq was performed using the weighted nearest neighbor method implemented in the R package Seurat 4.0. The ATAC library was normalized using a scaling factor calculated based on the total number of

cells in each cluster and the average sequencing depth for the same group of cells. Total counts at each base position for the track were divided by the scaling factor.

The transcription factor activity was assessed using the chromVAR package[26]. Briefly, the positional weight matrix was retrieved from the JASPAR CORE database (2020) and cell-type-specific chromVAR activities were calculated using the RunChromVAR wrapper in Signac. The differential transcription factor activity was computed using the FindMarkers function (FDR < 0.05). The motif enrichment analysis was performed on differentially accessible regions using the FindMotif function. The footrpinting analysis was performed using the Footprint function in Signac, and the footprinted motifs were plotted using the PlotFootprint function.

### Analysis of single-nuclei multiome sequencing from human ACDMPV and donor lungs

The single-nuclei multiome sequencing of ACDMPV patient lungs were previously reported and publicly available[34]. The raw sequencing reads from the ACDMPV dataset were mapped to the human reference genome hg38 (10x Genomics refdata-cellranger-arc-GRCh38-2020) using Cell Ranger ARC software. Nuclei passed the following criteria were included in downstream analysis: 1000–100,000 UMIs, <2 nucleosome signal values, ≥2 transcription start site (TSS) enrichment score, <5% reads overlapped with ENCODE hg38 blacklist regions. Nuclei that passed filters were used as input to Signac package for the subsequent bioinformatic analysis. Scrublet was applied to the RNA data to identify and remove potential doublets, and the SoupX package was used to remove technical ambient RNA counts from the RNA dataset. ATAC peaks were identified in qualified cell nuclei using MACS2 callpeak through the Signac CallPeaks function with default parameters. Peak-by-nuclei UMI matrix was generated using the FeatureMatrix function and normalized using the RunTFIDF function in Signac. Clusters based on chromatin accessibility patterns were identified using the Leiden algorithm. ACDMPV data were compared to donor multiome data (lung of 3-year-old healthy control) that were retrieved from the GEO database (GSE161383).

### Flow cytometry and FACS-sorting

Murine lungs were perfused through the right ventricle with 5 ml of PBS before removal from the thoracic cavity. The lungs were then minced into small pieces and enzymatically digested using 0.2 mg/ml Liberase TM (Roche, 05 401 127 001) and 100U/ml Deoxyribonuclease I (Sigma, DN25) as described[58,59]. Lung tissue homogenates were passed through nylon mesh (70 μm) with grinding force to obtain single-cell suspensions. Red blood cells were lysed in ammonium–chloride–potassium (ACK) lysis buffer. Cell concentrations were determined based on viable cell counts using a Countess II cell counter (Invitrogen, Carlsbad, CA). Subsequently, the single-cell suspensions were passed through cell strainer snap cap (Corning Life Sciences, 352235) mounted on round bottom polystyrene tubes to remove cell debris. Single-cell suspensions

were stained with fixable viability dye (Biolegend, 423108), incubated with TruStain FcX (Biolegend, 101320), and stained with a mixture of fluorochrome-conjugated antibodies as described. The following antibodies were used for flow cytometry: CD31 (48-0311-82, Thermofisher), CD45 (47-0451-82, Thermofisher), CD326 (17-5791-82, Thermofisher), CD140a (25-1401-82, Thermofisher), CD146 (134704, Biolegend) and CD106 (105719, Biolegend). The dilution for flow antibodies was 1:100. Data were acquired using BD LSR II cytometer equipped with FACSDiva 9.0 software. The compensation and data analyses were performed using FlowJo software (TreeStar version 10.8.1).

### Mouse embryonic stem cells and CRISPR-Cpf1 multiplex genome editing

The *Foxf1-GFP; CAG-tdTomato* mouse embryonic stem cell (ESC) line was generated and described previously[15]. CRISPR targeting constructs were generated by direct annealing of two complementary oligos into the pY095 plasmid (Addgene 84744) suitable for multiplex genomic fragment deletion. The oligo sequences targeting FEL1 and FEL4 are provided in the Supplementary table 2. The constructs were transfected into mouse ESCs with lipofectamine 2000, and cells were sorted into 96-well plates at a single-cell resolution. The cells were further cultured in 2i medium for clone selection, and single ESC clones were amplified for PCR genotyping. Mutant ESC clones were further validated using Sanger DNA sequencing.

### Mouse blastocyst complementation

Mouse CD1 females were super-ovulated by injections of serum gonadotropin and chorionic gonadotropin as described[60]. After mating, blastocysts were obtained at embryonic days (E) 3.5. For the generation of mouse chimeras, fifteen *Foxf1-GFP;CAG-tdTomato*-labeled wild type (WT) or mutant mouse ESCs were injected into the blastocysts using Piezo Micro Manipulator (Eppendorf PiezoXpert). After microinjection, blastocysts were transferred into pseudopregnant mouse females to undergo normal embryogenesis *in utero*. The chimeric mice were harvested at E18.5 for analysis.

### DNA reporter constructs and dual luciferase assay

FEL enhancers were amplified by polymerase chain reaction from mouse genomic DNA. Primers used for amplification of the enhancers and site-directed mutagenesis are provided in Supplementary Table 3. Briefly, a 5' primer (including a KpnI site), and a 3' primer (containing a NheI site), were used to amplify the DNA fragment. The amplified DNA products were purified and cloned into the KpnI-NheI linearized pGL4.25 reporter vector containing the minimal promoter to generate luciferase (LUC) reporter constructs. The putative FOXF1-binding sites and ETS-binding sites in FEL1 and FEL3 enhancers, the EBF1-binding site in the FEL4 enhancer and the GLI1-binding site in FEL2 were mutated using the site-directed mutagenesis. The constructs were verified by Sanger sequencing. The GLI1 expression plasmid was obtained from Addgene (catalog #62970). siRNA-mediated transient transfections and Real-time RT-PCR analysis were performed as previously described[15,58,60]. The dual luciferase assay was performed using the Dual-Luciferase Reporter Assay System (Promega) as described[9,61,62]. Luciferase assays were carried out in MFLM-91U cell line (Catalog Number: AMFLM-91U, Sevenhills bioreagents), an immortalized fetal endothelial cell line derived from mouse E19 lungs[63].

### Genomic analysis of distal *FOXF1* regulatory elements

The comparative genomic analysis of mouse *Foxf1* and human *FOXF1* distal regulatory elements was performed using the liftover tool from UCSC genome browser. Genome sequences were retrieved from the track hub "comparative genomics", and the DNA sequences in the track "Multiz Align" were retrieved as Mutation Annotation Format (MAF) and converted to a fasta format using the tool MAF to FASTA (one sequence per species, https://usegalaxy.org). The sequences were then aligned using the Clustal W2 algorithm implemented in UGENE toolkit[64]. The evolutionarily conserved DNA binding motifs in the accessible peaks from snATACseq datasets were first scanned in the CIS-BP (http://cisbp.ccbr.utoronto.ca) database, and the peaks were further validated with JASPAR database. Only motifs located in evolutionarily conserved DNA regions were considered as potential regulatory elements for further analysis. The syntenic chromosomes were retrieved using WashU epigenome browser.

### The single-cell RNA sequencing analysis of FEL1$^{-/-}$ and FEL4$^{-/-}$ cells derived from the mouse chimeric lungs via blastocyst complementation

The embryonic stem cell line of FEL1$^{-/-}$ and FEL4$^{-/-}$ were generated using CRISPR-Cpf1 duplex genomic editing. The chimeric lung cells used to prepare library were derived from mESCs via blastocyst complementation and the CD1 female adult mice were used for blastocyst complementation experiment. For each experiment, we used 4 mice to generate blastocyst complementation embryos. To prepare single-cell suspension, the lungs were finely minced into small pieces in RPMI-1640 medium containing 0.2 mg/ml Liberase TM (Roche) and 100U/ml Deoxyribonuclease I (Sigma) on ice. The minced lung pieces were incubated at 37 °C for 30 min. After enzymatic digestion, cell suspensions were forced to pass through a 70 μm cell strainer (Corning). Red blood cells were lysed in the Ammonium-Chloride-Potassium (ACK) lysis buffer (Thermofisher). Cell viability was determined in hemocytometer chambers using trypan blue stain. The chimeric lung cells derived from WT TdTomato+ ESCs were purified and used as a control. The chimeric lung cells derived from FEL1$^{-/-}$ and FEL4$^{-/-}$ TdTomato+ ESCs were purified using the same experimental conditions. Single-cell RNAseq libraries were generated using the GemCode Single-Cell Instrument and Single Cell 3' Library & Gel Bead Kit v3.1 and Chip Kit (10x Genomics) according to the manufacturer's protocol. Equal numbers of cells from each group (WT, FEL1$^{-/-}$ and FEL4$^{-/-}$) were used to prepare libraries. Approximately 11,000 cells were loaded for each sample with a targeted cell recovery estimate of 7000 cells. The libraries were sequenced using Novaseq 6000 at a sequencing depth of ~50,000 reads per cell. The raw BCL files directly obtained from Illumina sequencing, and cellranger mkfastq were used to convert sequencing reads into FASTQ format. Next, cellranger count from CellRanger software (v7.0.0, 10X Genomics) was used to generate gene-count matrices from the FASTQ files mapping to mm10 reference genome. Reads were processed by counting Unique Molecular Identifiers (UMI). Cells with unique RNA counts (nCount_RNA) less than 20,000 and more than 2000 were retained for downstream analysis. In addition, we also excluded cells that had more than 20% mitochondrial counts (mt.percent). For integrative analysis of cells from WT, FEL1$^{-/-}$ and FEL4$^{-/-}$ chimeric lungs, the three datasets were integrated by the Seurat software (version 4.0 in R4.1 environment). The data were log-normalized with a scale factor of 10000. The outlier cells were regressed out based on the percentage of mitochondrial reads and the number of UMIs based on distribribution of negative binomial model. Variable features were determined by the *FindVariableGenes* function in the Seurat package. After matrix normalization, *FindIntegrationAnchors* and *IntegrateData* functions were used to integrate the three datasets together. Next, the integrated matrix were used as input for linear dimensionality reduction via the principal component analysis (PCA) and the number of principal components used as input to UMAP projection were determined using the functions *ElbowPlot* and *DimHeatmap* in the Seurat package. The principal components with standard deviation > 3.5 were used as input for unsupervised clustering with a resolution set to 0.5 using the nonlinear UMAP down dimensional algorithm.

## Immunofluorescence of lung tissue sections

Cryosections from mouse lung tissue were prepared according to standard protocol as previously described[65]. Dissected mouse lungs were perfused with PBS, inflated, fixed with 3.7% formaldehyde and embedded into OCT on dry ice[66,67]. Lung cryosections from chimeric embryos were imaged for GFP and tdTomato using confocal microscope equipped with NIS-elements (version 5.25) software as described[68–70].

## Statistics and reproducibility

Statistical analyses were performed using R software and Graphpad Prism 9.0. One-way ANOVA and Student's *t* test were used to determine statistical significance. $P < 0.05$ was considered statistically significant. Multiple means were compared using one-way analysis of variance with the post-hoc Tukey's test when comparing the groups for which a post hoc analysis of each group was required. Non-parametric Mann–Whitney *U* test was used for datasets with n < 6 to determine statistical significance. Values were presented as mean ± standard deviation (SD). For luciferase experiment, we repeated three times for each experiment. For flow cytometry experiments, we repeated at least two times for each experiment. For immunofluorescence experiment, we checked results from six chimeric animals from blastocyst complementation experiment for each stem cell lines examined. No statistical method to predetermine sample size. We don't have any estimates of effect sizes used in this study. For cell experiments, flow cytometry experiments, animal experiments, and single-cell sequencing experiments, the investigators were blinded to group allocation during data collection and following data analysis. No statistical method was used to predetermine the sample size. No data were excluded from the analyses.

## Reporting summary

Further information on research design is available in the Nature Portfolio Reporting Summary linked to this article.

## Data availability

The single nuclei multiome data and single cell RNA seq data generated in this study have been deposited in the GEO database under accession code GSE217194. The single-cell RNA sequencing data from cells derived from WT and mutant mESCs via blastocyst complementation generated in this study have been deposited in the GEO database, and the single cell RNA sequencing dataset can be accessed at GSE217194. The single-cell ATAC sequencing data of mouse E8.5 embryo[25] are publicly available at GSE133244. The bulk-ATAC data for mouse embryonic lung development were generated by ENCODE consortium[30], and the processed files are available at GEO database with accession numbers GSE172744 (E14.5) [https://www.ncbi.nlm.nih.gov/geo/query/acc.cgi?acc=GSE172744], GSE172933 (E15.5) [https://www.ncbi.nlm.nih.gov/geo/query/acc.cgi?acc=GSE172933], GSE172813 (E16.5) [https://www.ncbi.nlm.nih.gov/geo/query/acc.cgi?acc=GSE172813] and GSE172769 (E18.5) [https://www.ncbi.nlm.nih.gov/geo/query/acc.cgi?acc=GSE172769]. The track data of phyloP and PhastCons measure evolutionary conservation and the multiple alignments of 60 vertebrate species based the mm10 reference mouse genome were downloaded from UCSC genome browser (http://hgdownload.soe.ucsc.edu/goldenPath/mm10/phyloP60way/ and http://hgdownload.soe.ucsc.edu/goldenPath/mm10/phastCons60way/). The human single-nuclei ATAC healthy control data[21] was deposited to the public GEO database with accession number GSE161383. The human ACDMPV patient multiome dataset published previously[34] was deposited at LungMAP Data Coordination Center (https://www.lungmap.net/omics/?experiment_id=LMEX0000004398). The FOXF1 Chipseq dataset[27] was deposited to the public GEO database under accession number GSE77951. The EP300 ChIPseq dataset from mouse fetal lung was generated by ENCODE consortium[30] and is publicly available in GEO database under accession number GSE91841. The EP300 ChIPseq data[29] from mouse adult lung is available in GEO under accession number GSE88789. Source data are provided with this paper.

## Code availability

Code used for the analysis of single cell multiome (ATAC + GEX) data is available in a public repository at https://github.com/WGLUN/Foxf1multiome.

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

## Acknowledgements

This work was supported by National Heart, Lung, and Blood Institute grants R01 HL141174 (VVK), R01 HL149631 (VVK), R01 HL152973 (VVK/TVK), R01 HL132849 (TVK) and R01 HL158659 (TVK).

## Author contributions

G.W. and V.V.K. designed the study; G.W., B.W., E.L., and Y.Z. conducted experiments; T.V.K. and V.V.K. provided funding for the study; G.W., M.G., T.V.K., J.A.W., and V.V.K. analyzed the data and provided critical insights; G.W. and V.V.K. wrote the manuscript with input from all authors.

## Competing interests

The authors declare no competing interests.
