## [Peer Review file · Nature Communications]

REVIEWER COMMENTS

Reviewer #1 (Remarks to the Author):

Heterozygous deletions and point mutations in the human FOXF1 gene locus are linked to Alveolar Capillary Dysplasia with Misalignment of Pulmonary Vein (ACDMPV). There is evidence that non-coding deletions in or near the FOXF1 gene locus cause human ACDMPV. Using multiome single-nuclei RNA and ATAC sequencing of mouse and human ACDMPV lungs, Wang et al. identified four conserved endothelial- and mesenchymal-specific enhancers in the FOXF1 gene locus. Furthermore, the authors show that endothelial FOXF1 enhancers are autoactivated and that mesenchymal FOXF1 enhancers are regulated by EBF1 and GLI transcription factors. In vivo tracing of ESCs harboring mutated FOXF1 enhancers in mouse embryonic lung tissue using blastocyst complementation validated the cell-specificity of these enhancers.

Overall, the manuscript is well-written, and the results provided are novel and interesting findings on the pathogenesis of ACDMPV. The authors could improve the manuscript by addressing the following points.

Major comments:

1. It is unclear why E18.5 was chosen to perform single nuclei RNA and ATAC sequencing using Foxf1-GFP positive cells isolated from mouse lungs. The authors previously demonstrated that the Foxf1-GFP line can recapitulate the endogenous Foxf1 expression in the mesenchyme earlier than E18.5 (Reference #7).
2. The lung-specific FOXF1 enhancer is known to be located ~270 kb upstream of the FOXF1 gene, as described in a recent review article (PMID: 33513839), as well as in the discussion, and the previously reported studies need to be described in the results to compare with the endothelial and mesenchymal enhancers identified in this study.
3. There is no description of the cells used for the luciferase assays in Fig. 4c, d. If primary vascular endothelial cells were used, it would be important to test whether knockdown of endogenous FOXF1 can lead to a reduction in the enhancer activity of FEL1 and FEL3. Furthermore, to validate the importance of the ETS binding sites in both FEL1 and FEL3, the combinatorial effects of the FOX:ETS motifs should also be investigated in vitro by luciferase assays.
4. The cells used for the luciferase assay in Fig. 5d should be described. As described above, do Ebf1 knockdown cells show reduced activity of the FEL4 enhancer? While recent evidence indicates that Ebf1

functions in pericyte cell commitment in vitro (PMID: 34272603), it is unclear whether mice deficient for Ebf1 exhibit defective lungs as seen in Foxf1 mutant mice.

5. The significance of the GLI-binding site in the FEL2 enhancer should be confirmed by in vitro studies like Fig. 5d.

6. Among the 11 cases that had deletions located upstream of the FOXF1 locus, only one patient (patient #2) was used for the multiome analysis (Fig. 6). Considering the possibility of phenotypic variations among the patients with deletions, at least one (or two) more patient(s) should be additionally analyzed.

7. The rationale for choosing FEL1 and FEL4 for functional validation (Fig. 7) with blastocyst complementation is not clearly described. As endothelial FEL1 and mesenchymal FEL2 enhancers are closely located and could be deleted in some ACDMPV patients, what would happen if both are deleted? Additionally, it is unclear why blastocyst complementation was performed using mESC-FEL1^{-/-} and mESC-FEL4^{-/-} lines to validate these enhancers. While this approach allows for assessing the effects of these deletions on the number of FOXF1-GFP positive cells in each cell type (Fig. 7e), Such analysis may not truly elucidate the functionality of the enhancers in vivo. In other words, mESC-FEL1^{-/-} and mESC-FEL4^{-/-} lines can be used to directly generate and analyze enhancer-specific mutant mice for FEL1 and FEL4, respectively.

Minor comments:

1. The title does not reflect the content of the manuscript well. The title should more accurately reflect that the authors identified endothelial- and mesenchymal-specific FOXF1 enhancers.

2. The discussion is somewhat brief. For example, there is no detailed discussion about how the four cell-type specific enhancers identified in this study are coordinately activated to regulate FOXF1 expression in the lung.

Reviewer #2 (Remarks to the Author):

General Comments:

This is an important study with excellent scientific rigor and the inclusion of blastocyst complementation studies. However, several results outlined below need further experiments to explain or validate the very interesting findings.

Specific Comments:

Abstract: The last line of the manuscript doesn't tie into the first and it should. This paper does help resolve why frequent non-coding FOXF1 deletions that interfere with endothelial and/or mesenchymal enhancers can lead to ACDMPV. Coming back to the clinical relevance will increase the impact of this important study.

Introduction: same comment, do the deletions or variants in non-coding DNA that are associated with ACDMPV affect these regulatory elements? How many of the non-coding variants intersect with these elements?

Results:

Line 182: Which ETS binding site is interacting with FOXF1 at the enhancer site, ERG, FLI, ETV?

Line 206: The authors should show whether GLI interacts with FEL 4 in a mesenchymal cell, i.e., pericyte, fibroblast or SMC as they did for EBL in pericytes

Line 226: The sentence that the deletion involved three different enhancers, one endothelial specific and the other two mesenchymal specific and all three reduced chromatin accessibility of the endothelial FEL1 does not seem to be substantiated by experimental data. Two enhancers are not active in endothelial cells, so perhaps it is just loss of the other endothelial enhancer, FEL3, that is important in

reducing FEL1 chromatin accessibility. Does deletion of each enhancer affect endogenous FOXF1 mRNA expression in the relevant cell type?

Figure 6: Why do the donor cells show accessibility of FEL2 in endothelial cells and not mesenchymal cells?

Line 256-Line 260: although not sorted by a specific marker can the authors comment on SMC numbers based upon ACTA2 or TAGLN staining and position, i.e., periarterial. Were the walls of the vessels thin or aneurysmal? How do the authors propose that loss of EC affected mesenchymal cells (pericytes, fibroblasts). Are there clues in the transcriptomic data that could be investigated to explain this interesting observation?

Methods:

Line 317: Details should be given about Male: Female distribution and whether sex was considered in analysis of the results.

Reviewer #3 (Remarks to the Author):

Wang et al. reported a study of the enhancers of Foxf1 in mice that follows up the study of Guo M et al (ref 28). Utilizing a single-cell RNA+ATAC joint assay (capturing 6000 cells/nuclei) on mouse embryonic lungs, the authors discerned relevant cell type-specific cis-regulatory enhancers of Foxf1 and their trans-acting regulators. They compared the enhancer locations with the human ACDMPV deletion locations and assessed for human chromatin accessibility using public single-cell multiomic data. Finally, using blastocyst complementation in mice, the authors revealed the essential roles of two enhancers (FEL1 and FEL4) across multiple lineages in fetal lung development.

The wealth of data from the single-cell multi-omics and blastocyst complementation studies stands as a significant contribution. However, the analytic approach and resultant conclusions require further refinement.

Major:

1. Was the mouse single-cell multi-omic data from one single library? Ideally, the authors need to show some levels of reproducibility by demonstrating comparable results from at least two replicates.
2. The cell type delineation in Fig 1b raises questions. Can the authors pinpoint airway/vascular smooth muscle cells? What are the subtypes in CAP1 and Matrixfibroblast? The authors may consider subclustering these clusters that have irregular shapes / branches, possibly also facilitated by label transfer using published mouse embryonic lung scRNA-seq data to better define cell types.
3. Similarly, Fig 6c's annotations appear overly simplistic, potentially skewing interpretations related to cell type-specific accessibility.
4. Related to the third point, Supplemental Fig 7b's reliance on the scores of merely two genes to ascertain cell population abundance seems inadequate.
5. While the conclusion regarding FEL1 and FEL4's roles in differentiation is based on FACS markers-defined broad lineage differentials, it does not preclude the possibility that these enhancers may also govern lineage proliferation and survival. The Guo M et al. study described abundance changes but they also demonstrated a blockage of differentiation showing increased progenitors and decreased committed cells. This, again, underscores the importance of granular cell population insights. Ideally, assessing the final chimeric lungs via scRNA-seq assays would provide insights into differentiation trajectory alterations.
6. A comprehensive comparative analysis between human and mouse single-cell multi-omic datasets is conspicuously absent. How do the cell type-specific peaks in proximity to human FOXF1 contrast with those in mice?

Minor:

1. Line 76: "studied" -> "studies"
2. Line 96: "identities" -> "identifies"
3. The term "down-dimension UMAP clustering" is a very confusing phrase. Dimension reduction and unsupervised clustering are two separate components of data processing. A more proper phrase can be "unsupervised clustering and UMAP embedding".
4. Fig 5c should indicate the percentage of cells that are double-positive
5. Clarity is required when referencing human FOXF1 versus mouse Foxf1. The font consistency needs attention, evident from "FOXF1 promoter" (Line 290) and "FOXF1 gene locus" (Line 296). Please rectify.
6. If the authors utilize "human donor lungs" as a standard control, consider specifying them as "healthy human donor lungs."

Reviewer #1 (Remarks to the Author):

Heterozygous deletions and point mutations in the human FOXF1 gene locus are linked to Alveolar Capillary Dysplasia with Misalignment of Pulmonary Vein (ACDMPV). There is evidence that non-coding deletions in or near the FOXF1 gene locus cause human ACDMPV. Using multiome single-nuclei RNA and ATAC sequencing of mouse and human ACDMPV lungs, Wang et al. identified four conserved endothelial- and mesenchymal-specific enhancers in the FOXF1 gene locus. Furthermore, the authors show that endothelial FOXF1 enhancers are autoactivated and that mesenchymal FOXF1 enhancers are regulated by EBF1 and GLI transcription factors. In vivo tracing of ESCs harboring mutated FOXF1 enhancers in mouse embryonic lung tissue using blastocyst complementation validated the cell-specificity of these enhancers.

Overall, the manuscript is well-written, and the results provided are novel and interesting findings on the pathogenesis of ACDMPV. The authors could improve the manuscript by addressing the following points.

We would like to thank the Reviewer for positive comments and valuable suggestions to improve our manuscript.

Major comments:

1. It is unclear why E18.5 was chosen to perform single nuclei RNA and ATAC sequencing using Foxf1-GFP positive cells isolated from mouse lungs. The authors previously demonstrated that the Foxf1-GFP line can recapitulate the endogenous Foxf1 expression in the mesenchyme earlier than E18.5 (Reference #7).

We apologize for the lack of rationale for these experiments. Previous studies demonstrated that FOXF1 is required for the formation of alveolar capillaries that consist of two endothelial cell types: CAP1 (gCAP, also known as general capillary cells) and CAP2 (aCAP, also known as aerocytes). Since differentiation of aCAP cells occurs around E17.5-E18.5 and continues after birth in the mouse lung, we have chosen E18.5 for our single-nuclei RNA and ATAC sequencing. We have modified the Results section of our manuscript to provide the rationale of choosing E18.5 (page 6).

2. The lung-specific FOXF1 enhancer is known to be located ~270 kb upstream of the FOXF1 gene, as described in a recent review article (PMID: 33513839), as well as in the discussion, and the previously reported studies need to be described in the results to compare with the endothelial and mesenchymal enhancers identified in this study.

We agree and apologize for our oversight. This review article has been added to references in the revised manuscript (new reference # 42). As requested by the Reviewer, we modified the Results and Discussion sections to compare the previously published 270kb upstream FOXF1 regulatory region with endothelial and mesenchymal enhancers identified in the present study (pages 13 and 17). We also provided a diagram to compare the 270kb upstream FOXF1 regulatory region with newly identified FEL1, 2, 3 and 4 endothelial and mesenchymal FOXF1 enhancers (Suppl. Fig. 10c).

3. There is no description of the cells used for the luciferase assays in Fig. 4c, d. If primary vascular endothelial cells were used, it would be important to test whether knockdown of endogenous FOXF1 can lead to a reduction in the enhancer activity of FEL1 and FEL3. Furthermore, to validate the importance of the ETS binding sites in both

FEL1 and FEL3, the combinatorial effects of the FOX:ETS motifs should also be investigated in vitro by luciferase assays.

We agree. As requested by the Reviewer, we provided this information in the revised manuscript (page 26). Luciferase assays were carried out in MFLM-91U cell line, which is an immortalized fetal endothelial cell line derived from mouse E19 lungs. MFLM-91U cells express endothelial genes, such as CD31/Pecam1, Flk1, CD34 etc, and represent a good model of fetal lung endothelial cells.

Furthermore, as requested by the Reviewer, we generated new LUC reporter constructs with FEL1 and FEL3 enhancers in which both FOX and ETS sites were disrupted using site-directed mutagenesis. The oligos used for mutations were provided in Suppl. Table 3. Using these double mutant LUC constructs, we performed additional co-transfection experiments to demonstrate that FOXF1 cooperates with ETS transcription factors FLI1 and ERG to activate FEL1 and FEL3 endothelial enhancers (new Supplemental Fig. 8a-c). These new data were incorporated into the revised manuscript (page 11). Therefore, evolutionarily conserved FOX:ETS motifs contribute to the regulation of FEL1 and FEL3 enhancers in the *Foxf1* promoter.

4. The cells used for the luciferase assay in Fig. 5d should be described. As described above, do Ebf1 knockdown cells show reduced activity of the FEL4 enhancer? While recent evidence indicates that Ebf1 functions in pericyte cell commitment in vitro (PMID: 34272603), it is unclear whether mice deficient for Ebf1 exhibit defective lungs as seen in Foxf1 mutant mice.

We agree. We performed the siRNA-mediated inhibition of Ebf1 in vitro to test if FEL4 activity was affected. A 50% decrease of Ebf1 expression was insufficient to inhibit *Foxf1* mRNA (measured by qRT-PCR) or FEL4 enhancer activity (measured by LUC assay) in cultured cells *in vitro* (new Supplemental Fig. 9a-b). We cannot achieve the higher percentage of Ebf1 inhibition in vitro due to the growth arrest. We modified the Results section of the revised manuscript to incorporate these new data (page 12).

We have also incorporated new Ebf1 pericyte references in the revised manuscript (new references # 48,50), and provided additional information about EBF1 functions in the Discussion section (page 18). Interestingly, the global *Ebf1* gene disruption did not cause lung defects in homozygous mutant mice (new reference # 49), probably, due to compensatory effects from other transcription factors.

5. The significance of the GLI-binding site in the FEL2 enhancer should be confirmed by in vitro studies like Fig. 5d.

We agree. As the Reviewer requested, we mutated the GLI-binding site in FEL2 enhancer and generated the mutant FEL2-LUC reporter construct. Disruption of the GLI-binding site significantly reduced the FEL2 enhancer activity in co-transfection experiments with GLI1 expression plasmid. These new findings are provided in new Supplemental Fig. 9c-d and incorporated into the Results section of the revised manuscript (pages 12).

6. Among the 11 cases that had deletions located upstream of the FOXF1 locus, only one patient (patient #2) was used for the multiome analysis (Fig. 6). Considering the possibility of phenotypic variations among the patients with deletions, at least one (or two) more patient(s) should be additionally analyzed.

We agree. As the Reviewer requested, we have now 2 ACDMPV patients sequenced: patient #1 (P1; new data) and patient #2 (P2; old data). In new patient P1, all FEL enhancers (FEL1-4) are deleted on one chromosome 16. These data are provided in new Suppl. Fig. 11b. Based on unsupervised clustering and UMAP embedding, pulmonary cells from both P1 and P2 ACDMPV patients aligned well (new Supplemental Fig. 11a). We modified the Results section to incorporate these new data (pages 13).

7. The rationale for choosing FEL1 and FEL4 for functional validation (Fig. 7) with blastocyst complementation is not clearly described. As endothelial FEL1 and mesenchymal FEL2 enhancers are closely located and could be deleted in some ACDMPV patients, what would happen if both are deleted? Additionally, it is unclear why blastocyst complementation was performed using mESC-FEL1^{-/-} and mESC-FEL4^{-/-} lines to validate these enhancers. While this approach allows for assessing the effects of these deletions on the number of FOXF1-GFP positive cells in each cell type (Fig. 7e), Such analysis may not truly elucidate the functionality of the enhancers *in vivo*. In other words, mESC-FEL1^{-/-} and mESC-FEL4^{-/-} lines can be used to directly generate and analyze enhancer-specific mutant mice for FEL1 and FEL4, respectively.

We agree that endothelial FEL1 and mesenchymal FEL2 enhancers are closely located and could be deleted in some ACDMPV patients. However, we don't have ACDMPV patient samples with FEL1/FEL2 double deletion without affecting other *FOXF1* FEL enhancers, limiting the use of human samples to interrogate the activity of FEL1 and FEL2. Since ACDMPV is an ultra-rare congenital disease, it is unlikely we can get such samples in near future. However, we have added a new multiome dataset from the available human ACDMPV sample with the deletion of all 4 FEL enhancers to the revised manuscript (new Suppl. Fig. 11a-d).

We also agree that blastocyst complementation allows for assessing the effects of FEL deletions on the number of FOXF1-GFP positive cells and their gene expression in each cell type. However, this analysis may not truly elucidate the functionality of the enhancers *in vivo*. We have added this statement to the Discussion section among limitations of our study (page 19).

We agree that the generation and analysis of stable mouse lines with homozygous FEL1^{-/-} and FEL4^{-/-} mutations will be ideal. Based on traditional methods, generation and analysis of new knockout mice from available mESC-FEL1^{-/-} and mESC-FEL4^{-/-} cell lines will require 1-2 additional years if we count (1) blastocyst injections of mESCs to generate mosaic mice with high contribution from mESCs, (2) the crossing of mosaic mice with WT mice to generate heterozygous lines with stable germ-line transmission, (3) the breeding of +/- mice to generate homozygous mice or embryos, and (4) proper and detailed analyses of -/- phenotypes. To save a time, we have tried to use an alternative approach, in which CRISPR/Cas9 was used to disrupt FEL1 and FEL4 in oocytes. This technology allows the generation of homozygous embryos in a single step. However, this method does not work well for genes with severe embryonic phenotypes, such as *Foxf1*, because homozygous embryos resorb in utero and haploinsufficient embryos can have developmental phenotypes. After 6 months of continuous attempts to generate FEL1^{-/-} and FEL4^{-/-} mice using several different CRISPR guides, we have only mosaic but not global -/- mice. This can be due to technical difficulties with CRISPR/Cas9 genome editing or due to the importance of FEL1 and FEL4 in mouse embryonic development. We hope to answer this question in our future studies.

Minor comments:

1. The title does not reflect the content of the manuscript well. The title should more accurately reflect that the authors identified endothelial- and mesenchymal-specific FOXF1 enhancers.

We agree. We have changed the title from “A Combination of Multiomics with Blastocyst Complementation Identifies Genomic Enhancers Involved in Alveolar Capillary Dysplasia” to new title: “Identification of Endothelial and Mesenchymal FOXF1 Enhancers Involved in Alveolar Capillary Dysplasia”.

2. The discussion is somewhat brief. For example, there is no detailed discussion about how the four cell-type specific enhancers identified in this study are coordinately activated to regulate FOXF1 expression in the lung.

We agree. We have revised the Discussion section to make it more detailed and provide additional information. The changes include the detailed discussion about how the four cell-type specific enhancers (FELs) identified in our study are coordinately activated to regulate FOXF1 expression in the lung (pages 19-20). The discussion is also graphically supported by Suppl. Fig. 20.

Reviewer #2 (Remarks to the Author):

General Comments:

This is an important study with excellent scientific rigor and the inclusion of blastocyst complementation studies. However, several results outlined below need further experiments to explain or validate the very interesting findings.

We would like to thank the Reviewer for positive comments and valuable suggestions to improve our manuscript.

Specific Comments:

Abstract: The last line of the manuscript doesn't tie into the first and it should. This paper does help resolve why frequent non-coding FOXF1 deletions that interfere with endothelial and/or mesenchymal enhancers can lead to ACDMPV. Coming back to the clinical relevance will increase the impact of this important study.

We agree. As the Reviewer suggested, we have modified the abstract to highlight the clinical relevance of our manuscript as follows:

“...This study resolves an important clinical question why frequent non-coding FOXF1 deletions that interfere with endothelial and mesenchymal enhancers can lead to ACDMPV”

Introduction: same comment, do the deletions or variants in non-coding DNA that are associated with ACDMPV affect these regulatory elements? How many of the non-coding variants intersect with these elements?

We agree. We have clarified this important point in the Introduction section (page 5) and the Results section of the revised manuscript (page 13). Genomic deletions in ACDMPV patients often involve FEL regulatory elements. Specifically, we provide information about 12 ACDMPV cases with non-coding FOXF1 deletions intersecting with FEL1-4 regulatory elements (Suppl. Fig. 10 and Suppl. Table 1).

Results:

Line 182: Which ETS binding site is interacting with FOXF1 at the enhancer site, ERG, FLI, ETV?

We agree. We have performed additional co-transfection experiments to address this comment. New Suppl. Fig. 8a-c is provided to demonstrate that ETS transcription factors FLI and ERG cooperate with FOXF1 to activate endothelial FEL1 and FEL3 enhancers. These new data were incorporated into the revised manuscript (page 11).

Line 206: The authors should show whether GLI interacts with FEL 4 in a mesenchymal cell, i.e., pericyte, fibroblast or SMC as they did for EBL in pericytes.

We agree. As the Reviewer requested, we performed additional experiments to test the importance of GLI in regulation of mesenchymal FEL2 and FEL4 enhancers. New Suppl. Fig. 9c-d shows that overexpression of GLI1 increases the activity of the FEL2 enhancer which contains a putative GLI-binding motif. Disruption of the GLI site via site-directed mutagenesis decreases the ability of GLI1 to activate FEL2 (new Suppl. Fig. 9c-d). In contrast, overexpression of GLI1 does not change the activity of FEL4 which lacks GLI-binding motif (new Suppl. Fig. 9d). These new data were incorporated into the revised manuscript (page 12).

Line 226: The sentence that the deletion involved three different enhancers, one endothelial specific and the other two mesenchymal specific and all three reduced chromatin accessibility of the endothelial FEL1 does not seem to be substantiated by experimental data. Two enhancers are not active in endothelial cells, so perhaps it is just loss of the other endothelial enhancer, FEL3, that is important in reducing FEL1 chromatin accessibility. Does deletion of each enhancer affect endogenous FOXF1 mRNA expression in the relevant cell type?

We apologize for the lack of clarity. We have removed the statement in question from the revised manuscript.

Furthermore, to address whether deletion of FEL1 or FEL4 enhancers affects endogenous mouse *Foxf1* mRNA in relevant cell types, we performed new single-cell RNA sequencing experiments using FACS-sorted WT, FEL1^{-/-} and FEL3^{-/-} cells derived from mouse ESCs in chimeras. We provided 3 new single-cell RNAseq datasets (WT, MT1 and MT4) to demonstrate that homozygous deletion of FEL1 decreases endogenous *Foxf1* mRNA in pulmonary endothelial cells, whereas homozygous deletion of FEL4 abolishes the expression of endogenous *Foxf1* mRNAs in both lung pericytes and fibroblasts (new Fig. 8a-d). These new data provide further support to our hypothesis that FEL1 is endothelial *Foxf1* enhancer and FEL4 is mesenchymal *Foxf1* enhancer. These new data were incorporated into the revised manuscript (pages 15-16).

Figure 6: Why do the donor cells show accessibility of FEL2 in endothelial cells and not mesenchymal cells?

We agree that it is a very interesting question which can be related to interspecies differences between FOXF1 gene regulation in the mouse and human. We modified the Discussion section to acknowledge these differences among limitations of our study (pages 19-20). Also, we would like to point out that the GLI-binding site in FEL2 enhancer is located in the region of increased chromatin accessibility for both endothelium and fibroblasts in the human donor lung. Therefore, it is possible that FEL2 can be active in both endothelial and fibroblast cell lineages depending on timing of lung development or duration of therapeutic interventions in ACDMPV. We have

provided new Suppl. Fig. 13c to show the location of the GLI-binding site in the accessible chromatin.

Line 256-Line 260: although not sorted by a specific marker can the authors comment on SMC numbers based upon ACTA2 or TAGLN staining and position, i.e., periarterial. Were the walls of the vessels thin or aneurysmal? How do the authors propose that loss of EC affected mesenchymal cells (pericytes, fibroblasts). Are there clues in the transcriptomic data that could be investigated to explain this interesting observation?

We agree. To address this comment, we used FACS-sorted, ESC-derived cells to perform single-cell RNAseq analysis to examine the percentage of smooth muscle cells (SMC) and distinguish them from myofibroblasts based on multiple markers, such as *Acta2*, *Actg2*, *Tagln*, *Crip2*, and *Cdh11*. We provided new Suppl. Fig. 19a-d to show these data.

Mosaic (chimera) mice that are produced via blastocyst complementation and used in our studies allow FACS-sorting and single-cell RNAseq analysis of mutant cells. However, since the majority of cells in mosaic mice derive from endogenous ESCs and are wild type (WT), these mice cannot be used to analyze the phenotype, such as thickness of the walls of the vessels, and aneurysms. We have started to generate the mice with a global deletion of FEL1 or FEL4 enhancers using CRISPR/Cas9 technology, but the creation of these mouse lines and the phenotype analysis can take 1-2 additional years and will be a subject of future research. While the transcriptomic data in chimeras are very interesting, their significance is limited without proper analysis of phenotypes in FEL1 or FEL4 global knockout mice that do not exist yet.

Methods:

Line 317: Details should be given about Male: Female distribution and whether sex was considered in analysis of the results.

The embryonic lungs were pooled from 3 littermate embryos for FACS sorting and single-cell RNAseq analysis. Sex of embryos was not determined prior to combining the samples. We have included this information in the revised manuscript (page 21). Sex of ACDMPV patients is reported in our recent manuscript (reference # 34).

We would like to point out that accurate determination of sex in mouse E18.5 embryos is very difficult to achieve without special imaging equipment and methodology, limiting the opportunity to study sex as a biological variable. In human ACDMPV, both sexes are affected by the disease with equal severity.

Reviewer #3 (Remarks to the Author):

Wang et al. reported a study of the enhancers of Foxf1 in mice that follows up the study of Guo M et al (ref 28). Utilizing a single-cell RNA+ATAC joint assay (capturing 6000 cells/nuclei) on mouse embryonic lungs, the authors discerned relevant cell type-specific cis-regulatory enhancers of Foxf1 and their trans-acting regulators. They compared the enhancer locations with the human ACDMPV deletion locations and assessed for human chromatin accessibility using public single-cell multiomic data. Finally, using blastocyst complementation in mice, the authors revealed the essential roles of two enhancers (FEL1 and FEL4) across multiple lineages in fetal lung development. The wealth of data from the single-cell multi-omics and blastocyst complementation studies stands as a significant contribution. However, the analytic approach and resultant conclusions require further refinement.

We would like to thank the Reviewer for recognizing the significance of our studies and for valuable suggestions to improve our manuscript.

Major:

1. Was the mouse single-cell multi-omic data from one single library? Ideally, the authors need to show some levels of reproducibility by demonstrating comparable results from at least two replicates.

We agree. We have provided an additional multiome dataset from another ACDMPV patient (new Suppl. Fig.11). The results in two ACDMPV patient samples showed similar changes in both cell subtypes and chromatin accessibility.

To avoid variations in mouse data, we have pooled 3 embryos in the same sample and generated a library from FACS-sorted Foxf1-GFP positive cells. Furthermore, since we previously published a scRNAseq data using the similar protocol to purify Foxf1-GFP positive cells (Reference # 7), we aligned 2 datasets (new snRNAseq and published scRNAseq) and observed similar patterns in cell distribution between 2 RNA libraries. We provide new Suppl. Fig. 3a-e to show comparable results.

2. The cell type delineation in Fig 1b raises questions. Can the authors pinpoint airway/vascular smooth muscle cells? What are the subtypes in CAP1 and Matrixfibroblast? The authors may consider subclustering these clusters that have irregular shapes / branches, possibly also facilitated by label transfer using published mouse embryonic lung scRNA-seq data to better define cell types.

We would like to thank the Reviewer for pointing this out. As suggested by the Reviewer, we performed additional bioinformatic analyses to identify smooth muscle cells as well as subtypes of CAP1 and Matrix fibroblasts, and better-defined cell clusters based on cell markers published previously (references # 37) and suggested by the recent publication from LungMap (reference # 20 – Sun et al, Dev Cell). Specifically, we provided new Suppl. Fig. 19a-e which distinguishes between smooth muscle cells and myofibroblasts. We also provided new Suppl. Fig. 2a-b (sub-clustering of matrix fibroblasts) and new Suppl. Fig. 2c-e (sub-clustering of CAP1). These data were integrated into the Results section of the revised manuscript (pages 7 and 16).

3. Similarly, Fig 6c's annotations appear overly simplistic, potentially skewing interpretations related to cell type-specific accessibility.

We agree. As the Reviewer requested, we provided more complex cell clustering in Suppl. Fig. 12a.

In Fig. 6c-d, we just wanted to highlight the chromatin accessibility of FEL enhancers among main cell lineages with distinct developmental origins, such as endothelial, epithelial and mesenchymal cells, using multiome data from human ACDMPV lung. The chromatin accessibility in FEL enhancers is consistent with *FOXF1* RNA expression data.

4. Related to the third point, Supplemental Fig 7b's reliance on the scores of merely two genes to ascertain cell population abundance seems inadequate.

We agree. We revised this figure to provide additional gCap and aCap markers to better define these cell populations (new Suppl. Fig. 12b).

5. While the conclusion regarding FEL1 and FEL4's roles in differentiation is based on

FACS markers-defined broad lineage differentials, it does not preclude the possibility that these enhancers may also govern lineage proliferation and survival. The Guo M et al. study described abundance changes but they also demonstrated a blockage of differentiation showing increased progenitors and decreased committed cells. This, again, underscores the importance of granular cell population insights. Ideally, assessing the final chimeric lungs via scRNA-seq assays would provide insights into differentiation trajectory alterations.

We agree. As the Reviewer suggested, we performed additional experiments and provided 3 additional single-cell RNAseq datasets that assessed FACS-sorted ESC-derived WT, FEL1^{-/-} and FEL4^{-/-} cells from chimeric lungs. These data are provided in new Fig. 8 and new Suppl. Figs. 18 and 19. These new data were incorporated into the revised manuscript (page 16).

6. A comprehensive comparative analysis between human and mouse single-cell multi-omic datasets is conspicuously absent. How do the cell type-specific peaks in proximity to human FOXF1 contrast with those in mice?

We agree. To address this comment, we compared our mouse snATAC library with snATAC library from healthy human donor lungs. Genomic mapping and annotation enable the comparison of genome structures in different species. Thus, we compared the peaks in proximity of human *FOXF1* and mouse *Foxf1* based on synteny between human and mouse chromosomes for genes *BANP* (downstream of *FOXF1*) and *GINS2* (upstream of *FOXF1*). The data are provided in new Suppl. Fig. 14. These new data show similar chromatin accessibility in both mouse and human snATACseq datasets.

Minor:

1. Line 76: “studied” -> “studies”

We agree. We have corrected this mistake.

2. Line 96: “identies” -> “identifies”

We have corrected this mistake.

3. The term “down-dimension UMAP clustering” is a very confusing phrase. Dimension reduction and unsupervised clustering are two separate components of data processing. A more proper phrase can be “unsupervised clustering and UMAP embedding”.

We would like to thank the Reviewer for pointing out this mistake. The mistake was corrected to “unsupervised clustering and UMAP embedding”.

4. Fig 5c should indicate the percentage of cells that are double-positive

We agree. We have revised Fig. 5c and its figure legend to include the percentage of cells that are double-positive for Ebf1 and Foxf1.

5. Clarity is required when referencing human FOXF1 versus mouse Foxf1. The font consistency needs attention, evident from "FOXF1 promoter" (Line 290) and "FOXF1 gene locus" (Line 296). Please rectify.

We agree and apologize for our inconsistency when referencing human *FOXF1* versus mouse *Foxf1*. The mistakes have been corrected in the revised manuscript.

6. If the authors utilize "human donor lungs" as a standard control, consider specifying them as "healthy human donor lungs."

We agree. As suggested by the Reviewer, we have replaced "human donor lungs" with "healthy human donor lungs" (page 13).

REVIEWERS' COMMENTS

Reviewer #1 (Remarks to the Author):

Overall, most of the comments have been addressed adequately. The experiments that would have addressed the question about the functionality of the enhancers (FEL1 and FEL4) in vivo were attempted but not successful due to technical difficulties with CRISPR/Cas9 genome editing in oocytes and/or the importance of FEL1 and FEL4 in mouse embryonic development. However, the revised manuscript has significantly improved.

Reviewer #2 (Remarks to the Author):

No further comments The authors have been appropriately responsive to my suggestions

Reviewer #3 (Remarks to the Author):

The authors have addressed all the concerns except that they should fix the GitHub link which shows 404 error.